# A three-dimensional ratiometric sensing strategy on unimolecular fluorescence–thermally activated delayed fluorescence dual emission

Xuping Li[1], Gleb Baryshnikov [2,3], Chao Deng[4], Xiaoyan Bao[1], Bin Wu[1], Yunyun Zhou[1], Hans Ågren[2,5] & Liangliang Zhu [1]

Visualized sensing through fluorescence signals is a powerful method for chemical and physical detection. However, the utilization of fluorescent molecular probes still suffers from lack of precise signal self-calibration in practical use. Here we show that fluorescence and thermally activated delayed fluorescence can be simultaneously produced at the single-molecular level. The thermally activated delayed fluorescence serves as a sensing signal with its wavelength and lifetime both altered correlating to polarity, whereas the fluorescence always remains unchanged as an internal reference. Upon the establishment of a three-dimensional working curve upon the ratiometric wavelength and photoluminescence lifetime vs. polarity, disturbance factors during a relevant sensing process can be largely minimized by such a multiple self-calibration. This strategy was further applied into a precise detection of the microenvironmental polarity variation in complex phospholipid systems, towards providing new insights for convenient and accurate diagnosis of membrane lesions.

[1] State Key Laboratory of Molecular Engineering of Polymers, Department of Macromolecular Science, Fudan University, Shanghai 200438, China. [2] Division of Theoretical Chemistry and Biology School of Biotechnology, KTH Royal Institute of Technology, SE-10691 Stockholm, Sweden. [3] Department of Chemistry and Nanomaterials Science, Bogdan Khmelnitsky National University, Cherkasy 18031, Ukraine. [4] MOE Key Laboratory of Macromolecular Synthesis and Functionalization, Department of Polymer Science and Engineering, Zhejiang University, Hangzhou 310027, China. [5] Department of Physics and Astronomy, Uppsala University, Box 516SE-751 20 Uppsala, Sweden. Correspondence and requests for materials should be addressed to L.Z. (email: zhuliangliang@fudan.edu.cn)

Fluorescent sensors continue to attract attention on account of their visualization ability and high sensitivity in detecting signals[1–3]. They can be applied in local microenvironments for monitoring the behavior of surrounding targets in various chemical and biological processes[4,5], like charging cellular functional proteins[6] and membrane systems[7]. Although the past decades have witnessed a significant progress in design and development of typical donor–acceptor based π-structures with alterable fluorescence signals and ability to sense local environmental change[8], a large majority of these fluorescent sensors work simply through the responsive variation of a single emission signal[9]. In such cases, fluctuations in probe concentration or fluid property can easily cause erroneous signal readings during a practical sensing process[10]. Currently, different calibration methods, e.g., complex fluorescence with a dual-band inverse, have been developed for ratiometric calibration that can make it possible to avoid these disturbances to some degree[11,12]. However, the usage of fluorescence type-only sensors will also suffer from the overlap of background autofluorescence[13]. It is thus desirable to explore unique photoluminescent techniques or strategies to overcome these dilemmas.

A simultaneous change in emission wavelength and lifetime of photoluminescence (PL) from a molecular emitter may overcome the above-mentioned obstacles and generate a breakthrough in sensing from a multi-mode perspective. Meanwhile, a dual-emission characteristic can also be employed for ratiometric sensing, namely one emission providing an internal reference and the other acting as a sensing signal, on the basis of the spectral and temporal features. To fulfill these requirements, an organic emitter that can produce both a normal fluorescence (FL) and a thermally activated delayed fluorescence (TADF) maybe a favorable choice.

TADF emitters have been regarded as the third class of organic light emitting diode materials, as their harvest of both singlet and triplet excitons without noble metals facilitates smart molecular engineering with an improved internal quantum efficiency[14]. TADF has been observed from intramolecular charge transfer (ICT) systems with a thermally accessible gap between the singlet and triplet excited states, enabling efficient up-conversion via thermally assisted reverse intersystem crossing (RISC)[15,16]. As reported for TADF molecules[17,18], the energy difference between the singlet and triplet states ($\Delta E_{ST}$) can be minimized by adjusting the energy levels of the lowest locally excited triplet state ($^3$LE) and CT states (both $^1$CT and $^3$CT). In addition, the hybridization of LE and CT can be observed by increasing the effective conjugation length[19]. Therefore, with the control of the donor as well as the conjugation length through steric or substituent patterns, LE and CT emission can be established, which could guarantee a dual-emission characteristic at the single molecular level.

In this work, a polarity sensor for building a three-dimensional (3-D) ratiometrically luminescent sensing method (change of wavelength and lifetime ratio vs polarity) was presented. It is based on the fact that the solvatoluminescence of the sensor connects to the ICT character of a molecular structure and that it can be addressed from donor–acceptor molecular engineering. To achieve a balanced contribution of LE and CT in the excited state, we present a molecular strategy to regulate the electron-donating ability of the donor group and to adjust the effective conjugation length by introducing different substituents. We designed and synthesized a series of donor–acceptor–donor (D–A–D)-type molecules with an acceptor diphenyl sulphone (DPS) group linked to three different donors (5-nitroindole, 5-aminoindole, and 5-acetaminoindole (AMID)) (here named compounds 1~3, respectively). With mutual control studies of all these compounds, we eventually found that compound 3 can work as we envisioned. The introduction of an acetamino structure in 3

results in an FL emission from the local state, while a TADF emission is produced from the $^1$CT state by the RISC process. The wavelength and lifetime of the TADF signal changed along with environmental polarity, whereas the FL one stays insensitive, making it possible to establish a 3-D calibration method with ratiometric wavelength and lifetime vs polarity. This strategy can be further applied for a precise detection of the microenvironmental polarity variation in complex phospholipid systems both in vitro and in vivo.

## Results

**Design and construction of the donor-acceptor-donor systems.** The syntheses of D-A-D type compounds are described in Supplementary Fig. 1. The chemical structures, as well as the strategy for constructing such a sensor system, are depicted in Fig. 1. The optimized geometry and electron-density distribution of the frontier molecular orbitals of these compounds were investigated using density functional theory with the Gaussian 16 package at the B3LYP/6-31G(d) level. The highest occupied and lowest

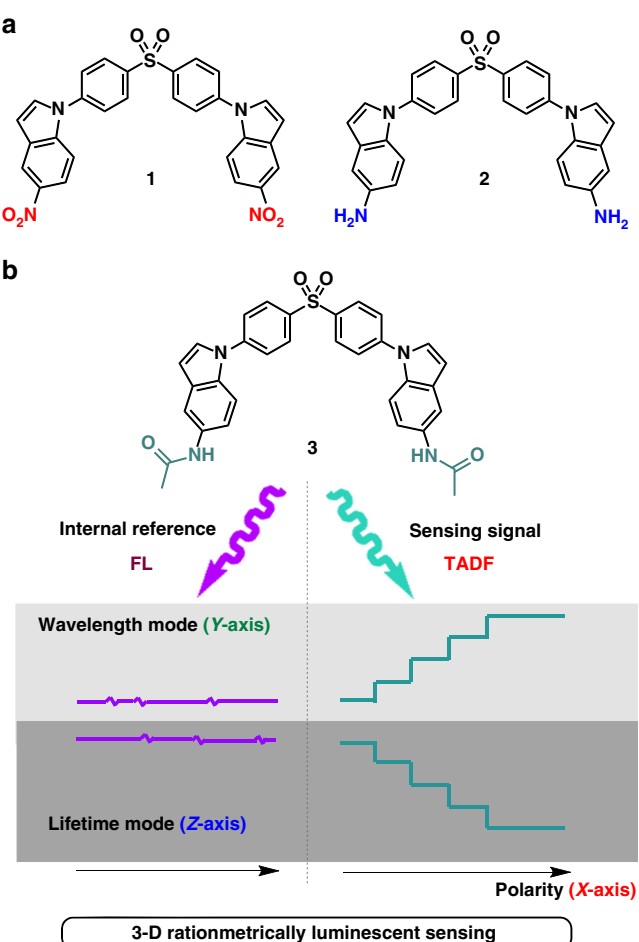

**Fig. 1** Chemical structures and the illustration of sensor working. **a** 1, 2, and **b** 3 and an outline for the construction of a 3-D ratiometric luminescent sensor (wavelength and lifetime ratio vs polarity), with the employment of the TADF from a CT state as a sensing signal, and the FL from the LE state as an internal reference. The illustration of the purple lines represents the FL emission staying insensitive to environmental polarity, while that of the dark cyanine lines represents the TADF bathochromically shifting in emission wavelength and reducing in lifetime along with the increase of polarity. CT, charge transfer; FL, fluorescence; TADF, thermally activated delayed fluorescence

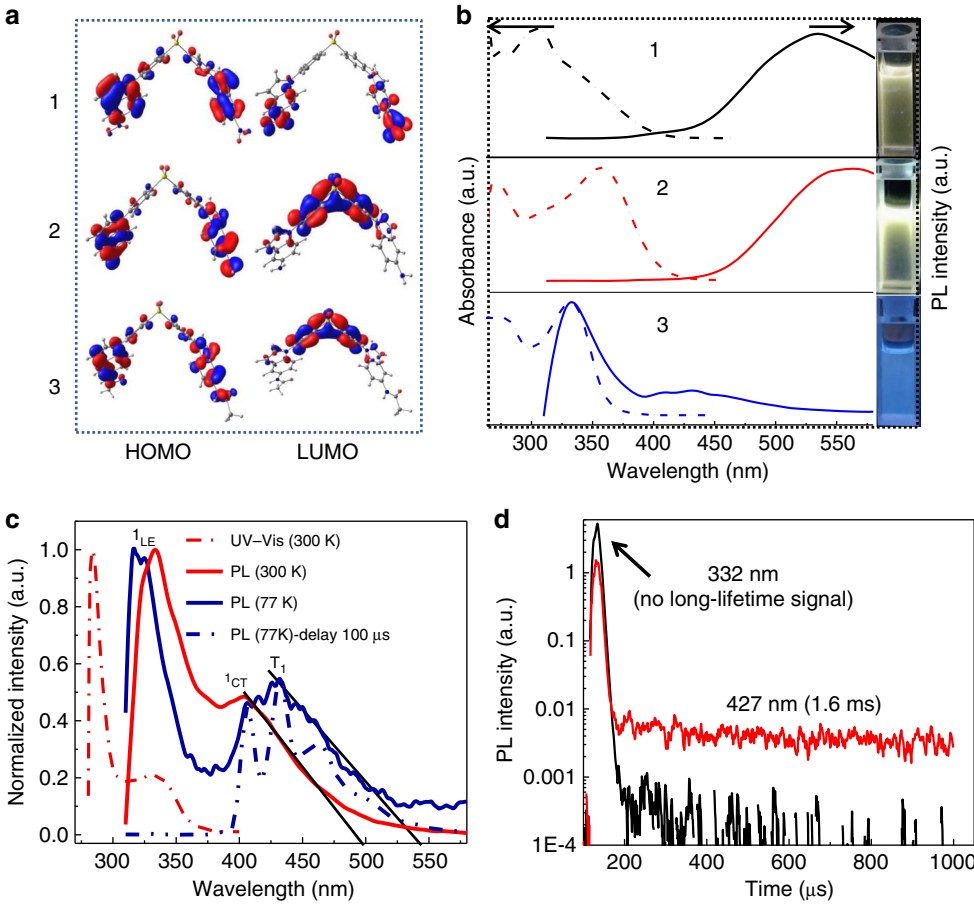

**Fig. 2** Photophysical study. **a** HOMO and LUMO of compounds 1, 2, and 3 calculated at the B3LYP/6-31G(d) level. **b** Absorption and emission spectra of the three compounds in DCM (2 μM) at room temperature. Inset: photographs under a UV light (365 nm). **c** UV/Vis absorption spectra at 300 K (red dash-dot curve), photoluminescence spectra at 300 K (red curve), at 77 K (blue curve), and photoluminescence spectra at 77 K with delay (blue dash-dot dot) of 3 in toluene (2 μM). **d** PL lifetime at 77 K. DCM, dichloromethane; HOMO, highest occupied molecular orbitals; LUMO, lowest unoccupied molecular orbitals; PL, photoluminescence

unoccupied molecular orbitals (HOMO and LUMO, respectively) are represented in Fig. 2a. The HOMO and LUMO of compound 2 and 3 are primarily localized over the electron-donating moiety and the acceptor moiety, respectively. Compound 1 shows an overlap on the phenyl bridge of the HOMO and LUMO orbitals because of the strong electron-withdrawing characteristic of the nitro groups resulting in LUMO localized on the nitrobenzene group. For 2, the introduction of amino groups on the indole unit enhances its electron-donating ability and consequently lowers the CT energy. From 2 to 3, an additional carbonyl is introduced to increase the LE component by extending the conjugate region [19], and the HOMO is delocalized over the whole backbone. Meanwhile, there is a small spatial overlap between HOMO and LUMO on the phenyl ring of the phenyl sulfone moiety. These results are well matched with the experimental values and can be further illustrated by the calculated data in Supplementary Table 1.

The photophysical properties of the three materials were investigated using absorption and PL spectroscopies to understand the properties of the excited states (Fig. 2b). Both 1 and 2 reveal a CT absorption band and broad emission spectra where the emission maximum of 2 (560 nm) is red shifted by about 30 nm from 1 (530 nm). Such a considerable PL red shift indicates that the strong electron-donating amino group can stabilize the HOMO[20]. Supplementary Fig. 2 shows the emission spectra of 1 and 2 in various solvents. The two molecules exhibit a clear

solvatoluminescent effect corresponding to the remarkable CT character of the excited state[21,22]. For 1, the emission demonstrates a relatively small red shift along with the solvent polarity increase due to the weak CT-state character from indole to nitrobenzene. The strengthened CT transition of 2, due to the very strong electron-donating ability of the amino group, leads to a large increase in the red shift of the emission spectra and an improvement of PL quantum yield (Supplementary Table 2).

Compared to 1 and 2, the PL spectra of 3 show obvious dual-emission bands at 332 and 435 nm. To further study this unique photophysical property, we collected the ultraviolet–visible (UV–Vis) absorption and PL spectra for 3 in toluene (Fig. 2c). The UV–Vis absorption spectra in toluene and dichloromethane (DCM) show a strong absorption peak at 284 nm, which can be attributed to the π–π* transitions of the AMID[23], and a broad absorption band at 309~376 nm. The absorption extinction coefficient (Supplementary Fig. 3a, b) of the band at 309~376 nm is $1.7 \times 10^4 \, M^{-1} \, cm^{-1}$, resulting from an efficient ICT between donor units and acceptor units. The absorption behaves similarly to the CT properties of some TADF molecules reported in literature[15] (Supplementary Fig. 3c). The PL spectrum of 3 in toluene gives a dual-emission characteristic, which is similar to the case in DCM. One emission band is narrow and strong with a maximum at 332 nm, originating from the π-conjugated LE state. The other emission is a broad band with a maximum at 401 nm, which can be ascribed to the ICT transition because of their

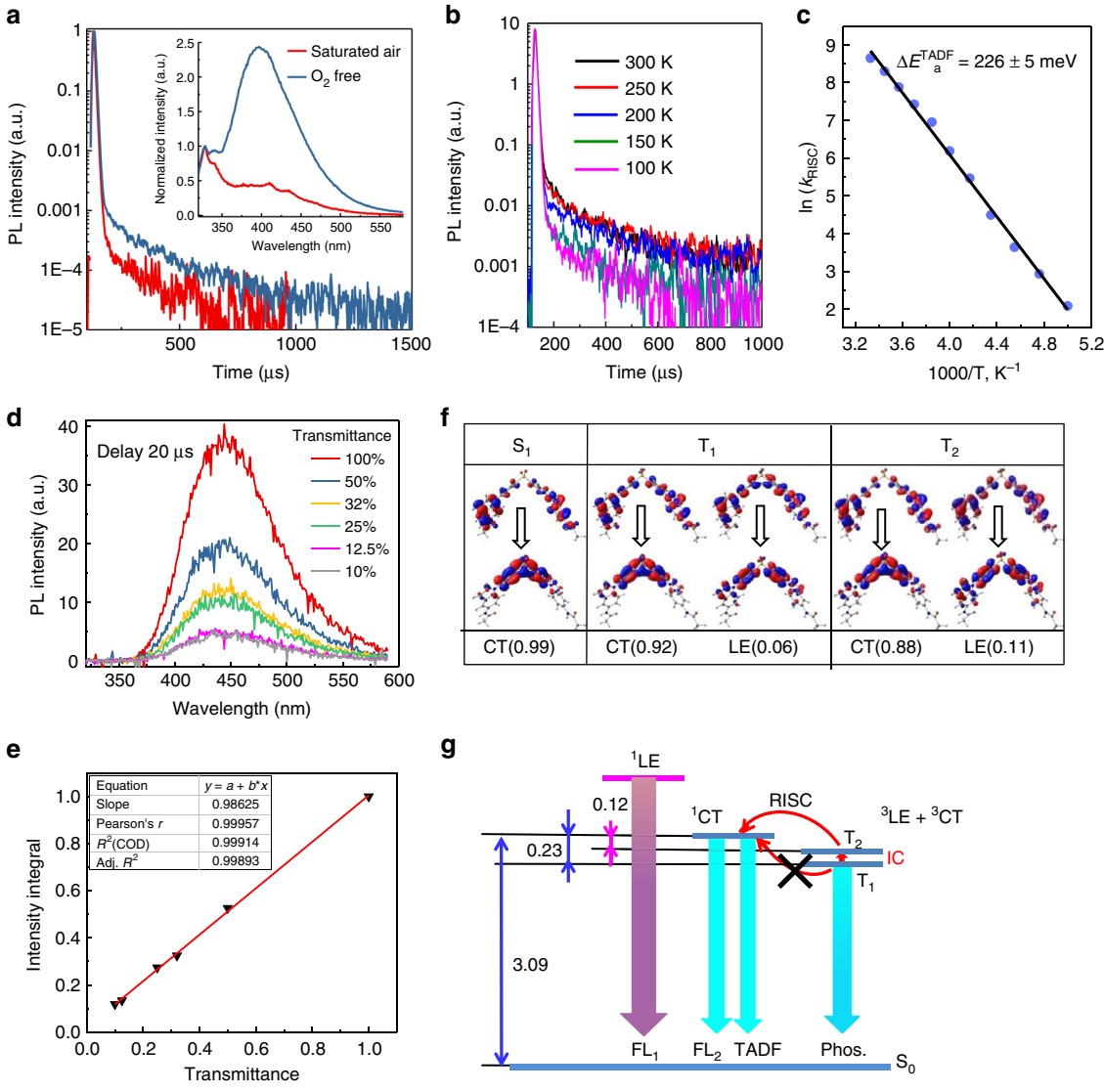

**Fig. 3** TADF characteristics. **a** Steady-state PL spectra (inset) and transient PL decay spectra in oxygen-free and saturated air conditions in toluene at 300 K. **b** Temperature-dependent PL lifetime measured from 100 to 300 K. **c** Arrhenius plots of the reverse ISC rate from the $T_1$ to the $S_1$ state measured for 6 wt% compound 3: DPEPO film under ambient condition. $k_{ISC}$ set to $5.5 \times 10^7$ s$^{-1}$. **d** Variation of intensity with excitation transmittance of compound 3 in DCM under oxygen-free condition (delay 20 µs). **e** A close linear dependence of intensity integral with excitation transmittance. **f** Pictorial representation of NTOs of the $S_1$, $T_1$, and $T_2$ states for 3. **g** Jablonski diagram for a proposed emissive mechanism of 3. Energy gaps in eV. DCM, dichloromethane; PL, photoluminescence; TADF, thermally activated delayed fluorescence

dipolar nature. These results can also be confirmed by the excitation spectra (Supplementary Fig. 3d) in DCM, in which the LE excitation band can be seen at 260~310 nm from the emission at 332 nm, while the ICT excitation band 310~370 nm is shown from the emission at 435 nm. The PL spectra also show dual emission at 332 and 427 nm at 77 K (Fig. 2c). The emission around 332 nm disappears in the time-resolved PL spectra with a 100 µs delay, while the emission around 427 nm shows a characteristic triplet excited state. Moreover, the lifetime measurement of the two emissions (Fig. 2d) shows that the former one is identified as fluorescence from the $^1$LE state because no long-lifetime signal can be detected. The latter, the one exhibiting a lifetime of 1.6 ms, indicates phosphorescence from its emissive $T_1$ excited state.

**TADF characteristic**. Based on the view that the long-lived $T_1$ excited state has been verified to exist, we aim next to clarify

whether the emission at 401 nm in toluene is TADF. We measured the steady-state PL spectra and transient PL decay spectra in oxygen-free and saturated air conditions in toluene (Fig. 3a). The luminescence intensity at 401 nm measured under $N_2$ atmosphere is much higher than that measured under saturated air conditions. Moreover, the lifetime of 401 nm under air atmosphere decreased from 167 µs, in the oxygen-free environment, to 55 µs. Thus, we can propose that the emission at 401 nm may be TADF from the $^1$CT state, which is obtained through an RISC process from the oxygen-sensitive triplet state. The TADF characteristic of the doped film of 3 was verified by analyzing the temperature-dependent PL lifetime (Fig. 3b). It appears in a two-component decay consisting of a nanosecond-order prompt fluorescence and a microsecond-order delayed fluorescence from the $^1$CT to the ground state (see also the proposed emission pathways in Fig. 3g). Emissions of the delayed components are gradually intensified when the temperature is increased from 100 to 300 K, confirming the existence of a thermal activation energy

barrier for TADF[24,25]. These observations unambiguously demonstrate that the RISC process for 3 was indeed accelerated by thermal energy and revealed a typical characteristic of TADF emitters[26]. The activation energy of the delayed fluorescence was estimated through an Arrhenius plot (Fig. 3c) for the dependence $k_{RISC}$ vs $T^{-1}$ (for $T = 200 \sim 300$ K, Supplementary Table 3), which was based on the well-known relationship[27]. In addition, a strictly linear dependence of the intensity integral on excitation transmittance is observed (Fig. 3d and Fig. 3e), confirming the pure thermally assisted nature of the TADF mechanism in compound 3.

To better understand these experimental results, natural transition orbitals (NTOs) were calculated to investigate the $S_0 \rightarrow T_1$ transitional character of 3 (Fig. 3f). The transition-responsible NTOs have a common non-zero area located on the benzene rings of DPS. An analysis of the NTOs demonstrates that the $T_1$ state of 3 can be assigned as a CT state with admixture of an LE component. The calculated $\Delta E_{ST}$ (energy gap between $^1$CT and $T_1$) for 3 equals 0.23 eV in a good agreement with calculated $\Delta E_a^{TADF}$ 0.226 eV and experimental value $\sim 0.17$ eV, deduced from the onsets of the fluorescence and phosphorescence spectra (Fig. 2c). Such a high $\Delta E_{ST}$ cannot provide an efficient direct $T_1 \rightarrow S_1$ RISC up-conversion; however, the next $T_2$ state is more appropriate for this process. The energy splitting between the $T_2$ and $^1$CT states is only 0.12 eV, which is usually sufficient for an efficient thermally activated population of the $^1$CT state by RISC. Coincidentally, we can see that the CT proportion decreases whereas the LE proportion increases for the $T_2$ state as judged by the NTOs. Therefore, the LE admixture is of great importance to the increase of spin-orbit coupling (SOC) between the $^1$CT ($S_1$) and $^3$LE + $^3$CT ($T_2$) states.

Figure 3 shows the most probable dual-emission mechanism. The excitation energy of 3 should be higher than that of its donor and acceptor analogs to ensure $^1$LE emission (fluorescence) of the donors. Because of the small $T_2 \rightarrow T_1$ gap (0.11 eV), the $T_2 \leftrightarrow T_1$ conversion should be fast and reachable by thermal activation under room-temperature conditions[28]. Moreover, the excitons in $T_2$ are then able to convert to the $^1$CT state by the RISC process and radiatively decay to the ground state as TADF. The $T_2 \leftrightarrow T_1 \rightarrow S_1$ RISC is much more efficient when compared with the $T_1$-$S_1$ RISC because of (1) a smaller $S_1$–$T_2$ energy gap and (2) the admixture of LE configuration in the $T_2$ state that provides efficient SOC between $T_2$ and $S_1$. Very recently, Monkman et. al[29] experimentally proved such a mechanism of reverse internal conversion delayed fluorescence for the well-known TCA blue emitter with a quite large $S_1$−$T_1$ gap (0.21 vs 0.17 eV for our case). They postulated that reverse internal conversion (rIC) between the $T_1$ and $T_2/T_3$ triplet states gives rise to the upper triplet-state population at room temperature, thus promoting an efficient RISC to $S_1$[29]. Also, we can expect in our case a strong RISC mechanism through the vibronic (nonadiabatic) coupling between the $T_2$ state, which contains the admixture of $^3$LE configuration, and the lowest charge transfer triplet state ($^3$CT)[30]. Such second-order coupling effects can also enhance the $k_{RISC}$ rate together with the RISC through the rIC step. For compound 2, the $T_1$ and $T_2$ states are almost pure CT states, and thus, there is no efficient second-order vibronic coupling between them and no significant SOC between them and the $S_1$ ($^1$CT) state. Thus, for compound 2, we observe only the prompt fluorescence, which is red shifted relative to compound 3 in agreement with time-dependent (TD)DFT calculations (2.78 eV for compound 2 comparing with the 3.09 eV for compound 3; Supplementary Table 1). Compound 1 is a standalone system for which the $S_1$, $T_1$, and $T_2$ states correspond purely to the LE nature. Moreover, the $T_1$ and $T_2$ states are degenerate and lie about 0.35 eV lower than $S_1$; so no

TADF is observed in this case. The unique TADF characteristic of compound 3 can be additionally explained by the dihedral angles between the donor and acceptor as shown in Supplementary Table 4. The excitation of compound 3 into $S_1$ and $T_1$ provides a dihedral angle with 48°[93°] and 38°[38°], indicating that the dihedral angle is changed only for one of the indole-based donor branches that do not strongly break the π conjugation between the second donor and acceptor parts. This fact explains that $^1$CT and $^3$CT emission is not quenched by the orthogonalization of donor and acceptor fragments upon excitation. However, it is important to note that due to the CT character of the $S_1$ and $T_1$ states for compound 2, the dihedral becomes 46°[90°] and 46°[85°] for $S_1$ and $T_1$ states, respectively, resulting in the quenching of its $T_1$ emission. For molecule 1, both $S_1$ and $T_1$ states are localized on the donor fragment, which has no impact on the target dihedral angle. These results are properly matched with the NTOs of compounds 1~3 (Supplementary Table 5).

**3-D ratiometrically luminescent sensing**. Based on the demonstration of the FL and TADF dual-emission characteristic of 3, we further found an interesting photophysical phenomenon regarding the experimental polarity. Figure 4a shows the PL measurements of 3, indicating that the FL signal, maximal at 332 nm, stayed without shift among different solvents, whereas the TADF band exhibited a remarkable red-shift feature along with the increase in solvent polarity from n-hexane (Hex), toluene, DCM, tetrahydrofuran (THF) to dimethyl formamide (DMF). Such a peculiar solvatoluminescence results from the fact that the FL signal from the π-conjugated LE state, which possesses a large orbital overlap between hole and electron wave functions[31], is insensitive to the solvent environment. In contrast, the TADF band from the CT excitation, showing a relatively small orbital overlap regarding the spatially hole and electron wave functions, is strongly affected by the polarity change. Figure 4b shows the log wavelength ratio of TADF towards FL in Hex, Toluene, DCM, THF, and DMF, respectively (refer to the curve across the green dots). With an increase in polarity from Hex (0.06) to DMF (6.4), the wavelength ratio of TADF to FL is progressively increased from 1.17 to 1.45, and the log value increased from 0.07 to 0.16. A linear relationship ($R^2 = 0.99$, Supplementary Fig. 4a) exists between the log value and the polarity, signifying that the wavelength can serve as a ratiometric luminescent sensor for polarity visualization.

On the other hand, the lifetime of FL and TADF also exhibits another distinct sensitivity mode toward polarity. Figure 4c shows that the lifetime of FL signal is always around 22 ns in different polarity solvents and the lifetime of the TADF band decreases from 55 μs in toluene to 29 μs in DCM, to 24 μs in THF and then to 1.6 μs in DMF. Although the TADF lifetime in Hex is not detected because of the faint emission, both the emission and the lifetime of phosphorescence can be observed in various solvents at 77 K (Supplementary Fig. 5). In this way, a linear relationship ($R^2 = 0.95$; Supplementary Fig. 4b) can also be drawn between the log lifetime ratio of TADF to FL and the polarity (Fig. 4b, the curve across the cyan dots). This calibration line enables an accurate quantitation of polarity from the relative lifetime of the sensors under various conditions.

Finally, we have now understood the overall molecular design represented by compound 3 with respect to ratiometric luminescent sensing. In view of taking full advantage of the wavelength and lifetime variation vs polarity, a 3-D ratiometrically luminescent sensing plot can be established based on two of the linear relationships on wavelength and lifetime modes of this molecule (Fig. 4b, the curve across the red dots), on which the FL signal provides an internal intensity reference and the

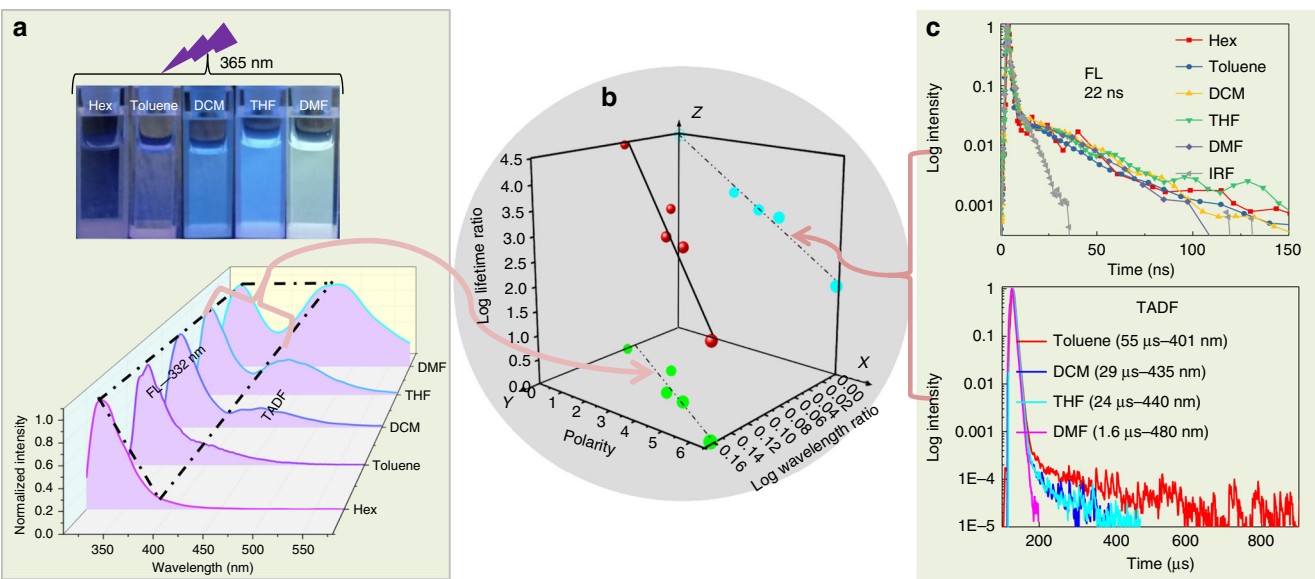

**Fig. 4** Correlation of wavelength and lifetime of TADF and FL with polarity. **a** Emission spectra of 3 in various solvents under ambient condition ($\lambda_{ex} = 300$ nm) and the photographs of 3 under UV light (365 nm). **b** Linear-fitting curves of the log value of wavelength and lifetime ratio (TADF to FL) with polarity change separately, as well as the formation of a 3-D plot diagram accordingly. **c** PL lifetime of FL and TADF bands in different solvents. FL, fluorescence; PL, photoluminescence; TADF, thermally activated delayed fluorescence

TADF one acts as a solvatochromic signal within the two modes. Such a system thus creates a multiple self-calibration and will undoubtedly minimize disturbance factors in different sensing situations thereof. We emphasize that these changes in photophysical behavior are really polarity dependent. Other luminescent pathways like exciplex emission can be excluded, since the dynamic light scattering analysis showed that there was no intermolecular interaction or aggregation effect of our molecule in these polarity environments. We studied the effects of different viscosities on steady-state PL and transient PL decay spectra of compound 3 by adding different glycerol concentrations to the ethanol. The results (Supplementary Fig. 6) show that although the intensity of the dual band can be affected by the increase in glycerol concentrations, the wavelength and lifetime vary only little. It suggests that the choice of wavelength and lifetime as the dimensions is practical. Another point worth mentioning is that all these wavelengths and lifetimes of the TADF band were investigated in an air-saturated environment, which is sufficient for the establishment and application of the 3-D plot diagram from the PL quantum yield results (Supplementary Table 6). This is because oxygen can quench TADF to some extent (Supplementary Fig. 7), but not completely. Therefore, the ratiometric PL sensing strategy demonstrated in this work is still valid and useful. This finding suggests that the employed luminescence signals are stable and that the usefulness of the parameters still holds in microenvironments in which oxygen is ubiquitous.

**Application in simulated membranes**. Phospholipids (PLs) are fundamental molecular components of a biological membrane and have been extensively used to determine various membrane properties. Cholesterol (Chol) plays a major role in the lateral organization of the lipid bilayer as it regulates membrane fluidity and permeability[32]. As an abnormal polarity alteration along with cholesterol-content change usually is linked to a membrane lesion, we expect that the 3-D ratiometric luminescent sensing strategy is applicable for the detection of the microenvironmental polarity change along with the alteration of the content of cholesterol in membranes. Herein, we utilized PLs systems to simulate the membranes, of which the polarity was changed by adding different Chol contents. Figure 5a gives a schematic illustration of a PLs system with different Chol contents and Fig. 5b shows the transmission electron microscopy (TEM) image of its self-assembled nanostructure with a micelle form.

We designed eight different Chol contents (mass ratio of 3/PLs/Chol: 1/100/0, 1/100/20, 1/100/40, 1/100/60, 1/100/80, 1/100/100, 1/100/150, and 1/100/200) for sensing in simulated membranes. The PL spectra and the lifetime of every Chol content were measured and are shown in Supplementary Fig. 8 and Supplementary Fig. 9, respectively (see also the corresponding value in Supplementary Table 7). The maximum emission wavelength blue shifts from 457 to 445 nm when the Chol content increases from 1/100/0 to 1/100/60, and red shifts from 445 to 466 nm when the Chol content increases from 1/100/60 to 1/100/100, and then remains stable when the Chol content increases from 1/100/100 to 1/100/200. Moreover, the lifetime increases from 4.12 to 9.09 µs when the Chol content increases from 1/100/0 to 1/100/60, and decreases from 9.09 to 3.17 µs when the Chol content increases from 1/100/60 to 1/100/100, and then remains stable when the Chol content increases from 1/100/100 to 1/100/20. Therefore, we divided the microenvironmental polarity of compound 3 in the complex PLs system into three stages: (1) starting value with a Chol content of 1/100/0; (2) polarity decreased from 1/100/0 to 1/100/60; (3) polarity increased from 1/100/60 to 1/100/100. Here, we analyzed three representative Chol contents of 0, 60, and 100 named A, B, and C in detail. When we put points A ($Y = \log (457/332)$, $Z = \log (4120/22)$), B ($Y = \log (445/332)$, $Z = \log (9090/22)$), and C ($Y = \log (466/332)$, $Z = \log (3170/22)$) in $YZ$ plane of the established 3-D plot diagram (Fig. 5c), the corresponding polarity value of PLs can be deduced by the 3-D working plot.

Thus, the measured values from A, B, and C points for polarity are 5.16, 4.41, and 5.71, respectively, by deduction from a linear-fitting curve of the wavelength ratio. Meantime, the values deduced from the linear-fitting curve of the lifetime ratio are 5.55, 4.67, and 5.85. As a result, we can reduce the measurement error by half in average to obtain the arithmetic mean of polarity; namely, it is about 5.355 in PLs without Chol (from point A).

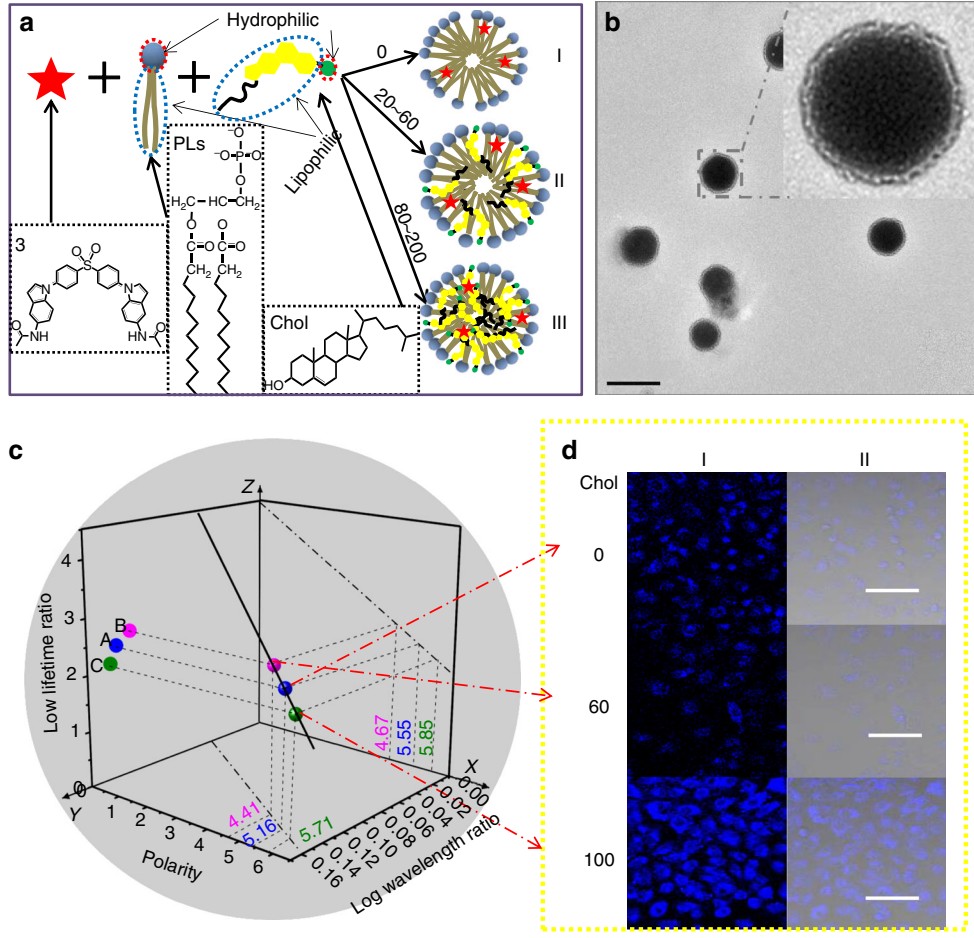

**Fig. 5** Application of the 3-D ratiometric luminescent sensing strategy in simulated membranes. **a** Schematic illustration of the complex PLs systems with different Chol contents. **b** TEM image of the PLs systems (inset: a zoom-in image). Scale bar, 200 nm. **c** 3-D plot diagram based on the single emitter 3. The points represent the luminescent behaviors with different Chol contents in the complex PLs system, thus for deducing the corresponding polarity values. **d** Confocal fluorescence microscopy imaging of Hela cells incubated with the complex PLs system with different Chol contents of 0, 40, and 100. I, luminescence imaging and Π, merged luminescence and bright-field imaging collected by the 4′,6-diamidino-2-phenylindole (DAPI) channel ($\lambda_{ex} =$ 408 nm). Scale bar, 50 μm. PLs, phospholipids; Chol, cholesterol

Similarly, the combination of PLs and Chol with a content of 60 caused a decreased polarity to be 4.54 (from point B), and the excessive Chol in PLs with content 100 resulted in an increased polarity (5.78, from point C). To further explain the change of polarity with differenet Chol contents, TEM and dynamic light scattering are shown in Supplementary Fig. 10. Without Chol in the membrane, the polarity 5.16 originates from the long alkyl chain of the PLs (Fig. 5a (I)), with a Z-average diameter of ca. 85 nm. When the mass ratio of 3/PLs/Chol is 1/100/60, the Chol molecules are arranged perpendicular to the surface (Fig. 5a (II))[32]. The central part of the membrane consists of the long alkyl chain of PLs and the polycyclic aromatic hydrocarbon chain of Chol; so its polarity decreases accordingly with the Z-average diameter increase of ca. 140 nm, indicating that more substances are formed in the membranes. However, the polarity clearly increases for the membrane with 3/PLs/Chol of 1/100/100 because (1) the sensor is largely exposed to the Chol environment and (2) the presence of a higher proportion of Chol may induce a wider separation of the phospholipid headgroups, which may better surround the sensor molecules (Fig. 5a (Ⅲ)). Moreover, the Z-average diameter decreased to ca. 120 nm, due to the condensing effect of Chol on PLs by reducing the average orientational and positional ordering of the alkyl chains[33]. These results also indicate that our sensor firstly probed the membrane polarity quantificationally and that

the application of such a multiple self-calibration will further reduce the maximum error between the experimental value and the real value, which features that the 3-D ratiometrically luminescent sensing method can reveal a higher accuracy and better avoid disturbance factors in detection, as compared with other fluorescence and ratiometric fluorescence probing methods[34].

In addition to the in vitro simulated study, compound 3 can also be used for cellular experiments. Supplementary Fig. 11 shows the viability of the Hela cells using four parallel CCK-8 experiments. No obvious cytotoxicity was observed at low concentrations from 0.125 to 4 μM after 24 h incubation, reflecting the low cellular toxicity of the sensor. Moreover, confocal fluorescence microscopy was carried out and the photo-stability of compound 3 is shown in Supplementary Fig. 12. Hela cells were grown under the complex PLs system with different Chol contents of 0, 60, and 100. As shown in Fig. 5d, opposed to the sample without Chol, cells incubated with the complex PLs system with a Chol content of 60 showed a relatively weak brightness in luminescence mode. However, the cells incubated with the PLs system with a Chol content of 100 showed a strong luminescence signal. The insensitivity of the signal to oxygen might be caused by the fact that phospholipids may effectively hinder the contact and collisions between the TADF molecule

and oxygen molecules. This factor makes our probe reliable in practical usage. Such a momentous phenomenon indicates a polarity change in agreement with the in vivo study by the visualization of the sensor strategy, implying that our established strategy has a good potential for the diagnosis of cholesterol-related membrane lesions.

## Discussion

In summary, we have demonstrated a fluorescence–thermally activated delayed fluorescence (FL–TADF) dual-emission strategy using a single molecular platform with hybridized local and CT excited states. Molecular engineering with an appropriate donor (AMID)-acceptor (DPS) linkage played a key role here. While an excellent TADF property was observed as a result of the fine-tuned energy of the $^1$CT and $^3$LE + $^3$CT states, the introduction of the LE component simultaneously caused a strong FL emission with a short lifetime. A 3-D ratiometric luminescent sensing method can be established since the wavelength and lifetime of the TADF signal both change along with polarity whereas the two features of the FL emission remain insensitive. Our present work provides a strategy to quantitatively study the polarity change in a simulated membrane system by monitoring the luminescent behavior with different Chol contents forthright in vitro and in vivo. Furthermore, we emphasize that this application is based on a well-established time-resolved sensing strategy and can pave a way to greatly reduce the measurement error and to avoid disturbance factors during practical sensing.

## Methods

**General**. 5-Nitroindole, potassium carbonate, bis($p$-fluorophenyl) sulfone, SnCl$_2$·2H$_2$O, and acetylchloride were purchased from Energy Chemical and used as received without further purification. $^1$H NMR and $^{13}$C NMR spectra were measured on a Bruker 400 L spectrometer in C$_2$D$_6$O$_6$ at room temperature. High-resolution mass spectrometry data were measured by Matrix Assisted Laser Desorption Ionization-Time of Flight/Time of Flight Mass Spectrometer (5800). Elementary analysis was performed on a Thermo Finnigan Flash EA1112. The high-performance liquid chromatography analysis was performed on an Agilent 1260 HPLC system. The mobile phase consisted of methanol and acetonitrile with the flow rate of 1 mL/min. The absorption wavelength used was set at 330 nm. Hundred percent acetonitrile was used as the running buffer.The UV–Vis absorption spectra were recorded on a Shimadzu 1800 spectrophotometer. The emission spectra, time-resolved emission spectra, and transient PL decay characteristics of the samples were recorded on QM40 (PTI, Horiba Scentific). Temperature was controlled by a cryostat (Advanced Research System Inc.). A nitrogen laser (PTI) with an excitation wavelength of 370 nm was used as the excitation source. The fluorescence quantum yields of solution were measured on QM40 with an integrating sphere (φ 150 mm).

**Bis[4-(5-nitroindole)phenyl] sulfone (1)**. 5-Nitroindole (4.86 g, 30 mmol) was added to a solution of potassium carbonate (19.35 g, 140 mmol) in dry 1-methyl-2-pyrrolidinone (NMP) (30 mL). After the solution was stirred at room temperature for 10 min, bis($p$-fluorophenyl) sulfone (3.56 g, 14 mmol) in dry NMP (20 mL) was added, and then the mixture was stirred at 140 °C for an additional 4 h. After cooling, the mixture was poured into 500 ml of water, and the celadon precipitate was filtered and rinsed with water and petroleum ether. Then it was dried under high vacuum (7.15 g, 95%). $^1$H NMR (400 MHz, DMSO-d$_6$) $\delta$ (ppm): 8.69 (d, $J =$ 2.3 Hz, 2H), 8.33–8.23 (m, 4H), 8.10 (dd, $J = 9.2$, 2.3 Hz, 2H), 8.03 (d, $J = 3.4$ Hz, 2H), 8.01–7.93 (m, 4H), 7.83 (d, $J = 9.2$ Hz, 2H), 7.07 (d, $J = 3.3$ Hz, 2H). $^{13}$C NMR (100 MHz, DMSO-d$_6$): $\delta =$ 142.80, 142.39, 139.45, 138.11, 132.62, 130.04, 129.56, 125.55, 118.46, 118.41, 111.86, 107.40; MALDI-TOF MS, $m/z$: [M + H]$^+$ 540.5. Elemental analysis (for C$_{28}$H$_{18}$N$_4$O$_6$S), calculated: C (62.45), H (3.37), N (10.40); found: C (62.38), H (3.40), N (10.45). The structural characterization is shown in Supplementary Fig. 13.

**Bis[4-(5-aminoindole)phenyl] sulfone (2)**. A mixture of bis[4-(5-nitroindole) phenyl] sulfone (5.38 g, 10 mmol) and SnCl$_2$·2H$_2$O (22.56 g, 100 mmol) in 300 mL of EtOH was heated to reflux for 20 h. After cooling, the mixture was poured into 500 mL of ice water and adjusted to pH 7 with NaOH. This mixture was then extracted with DCM (3 × 100 mL) and the extracts were dried (Na$_2$SO$_4$) and concentrated to be the yellow precipitate. Then it was dried under high vacuum (3.82 g, 80%). $^1$H NMR (400 MHz, DMSO-d$_6$) $\delta$ (ppm): 8.14–8.09 (m, 4H), 7.82 (dd, $J = 11.6$, 4.7 Hz, 4H), 7.59 (d, $J = 3.4$ Hz, 2H), 7.45 (d, $J = 8.8$ Hz, 2H), 6.79 (d, $J = 2.1$ Hz, 2H), 6.61 (dd, $J = 8.8$, 2.2 Hz, 2H), 6.51 (d, $J = 3.3$ Hz, 2H), 4.92

(s, 4H). $^{13}$C NMR (100 MHz, DMSO-d$_6$): $\delta =$ 144.17, 143.50, 137.25, 131.50, 129.71, 128.36, 128.02, 123.19, 113.24, 111.57, 105.13, 104.62; MALDI-TOF MS, $m/z$: [M + H]$^+$ 478.2. Elemental analysis (for C$_{28}$H$_{22}$N$_4$O$_2$S), calculated: C (70.27), H (4.63), N (11.71); found: C (70.05), H (4.65), N (11.68). The structural characterization is shown in Supplementary Fig. 13.

**Bis[4-(5-acetaminoindole)phenyl] sulfone (3)**. Bis[4-(5-aminoindole)phenyl] sulfone (0.96 g, 2 mmol) was dissolved in 8 mL DMF and then trimethylamine (0.15 mL) was added. Acetylchloride (0.28 mL, 4 mmol) was added dropwise in N$_2$ at 0–3 °C, and then the mixture was warmed to 40 °C for 8 h. After cooling, the mixture was poured into 50 ml of ice water, and the precipitate was filtered and rinsed with water and further purified by column chromatography (DCM: THF = 1:1, v/v) to afford a brown solid (0.88 g, 78%). $^1$H NMR (400 MHz, DMSO-d$_6$) $\delta$ (ppm): 9.94 (s, 2H), 8.18 (d, $J = 7.1$ Hz, 4H), 8.04 (s, 2H), 7.89 (d, $J = 7.1$ Hz, 4H), 7.75 (s, 2H), 7.65 (d, $J = 8.2$ Hz, 2H), 7.33 (d, $J = 8.1$ Hz, 2H), 6.74 (s, 2H), 2.06 (s, 6H). $^{13}$C NMR (100 MHz, DMSO-d$_6$): $\delta =$ 168.38, 143.78, 137.96, 133.87, 131.43, 130.26, 129.83, 129.08, 124.00, 116.38, 111.54, 111.23, 105.92, 24.43; MALDI-TOF MS, $m/z$: [M + Na]$^+$ 585.5. Elemental analysis (for C$_{32}$H$_{26}$N$_4$O$_4$S), calculated: C (68.31), H (4.66), N (9.96); found: C (68.12), H (4.68), N (9.93). The structural characterization is shown in Supplementary Fig. 13.

**Computational details**. The structure of molecules 1~3 was optimized in its ground state (S$_0$) at the DFT level of theory using the B3LYP[35] hybrid functional and 6-31 G(d)[36] basis sets. The excitation energies were calculated by employing the specially parameterized B3LYP functional ($a_X^{HF} = 0.12$, $a_X^{Slater} = 0.88$) with the 6-31 + G(d)[37] basis set within the TDDFT approach[38]. The exchange tuning of B3LYP functional requires the correct explanation of excited states that combine both CT and local excitation LE characters and possess wide "overlapping" area for the transition-responsible orbitals. We have also tested the dispersion-containing wB97XD functional and long-range corrected CAM-B3LYP functional for the optimization of the studied systems and the subsequent TDDFT calculations. We have found that there is no significant difference in the structure of S$_0$, S$_1$, and T$_1$ states compared with the B3LYP results. At the same time, the energies of excited states were found to be strongly overestimated when compared with the parameterized B3LYP functional. All the calculations were performed using the polarizable continuum model[39], taking the dielectric constant for DCM ($\varepsilon = 8.93$) as the reference. NTO analysis was performed to examine the nature of the excited states. All the calculations were performed within Gaussian 16 program package[40].

## Data availability

The data that support the findings of this study are available from the corresponding author upon reasonable request.

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

## Acknowledgements

This work was supported by NSFC/China (21628401, 21644005) and partially by the National Key Research and Development Program of China (2017YFA0207700). The calculations were performed with computational resources provided by the High Performance Computing Center North (HPC2N) in Umeå, Sweden, through the project "Multiphysics Modeling of Molecular Materials" (SNIC 2017-12-49). H.A. and G.B. acknowledge the Carl Tryggers foundation (Grant No. CTS 16:536 and 17:514). G.B. also thanks the Ministry of Education and Science of Ukraine (project number 0117U003908).

## Author contributions

L.Z. and X.L. conceived this project and designed the experiments. X.L. and B.W. did the synthetic work. X.L. and C.D. carried out the photophysical studies as well as the structural characterization. X.L. and X.B. carried out the cellular experiments. G.B. carried out the calculation studies. X.L., G.B., Y.Z., and H.A. contributed to data analysis. X.L. prepared the manuscript and the other authors helped revising the paper.

## Additional information

**Competing interests:** The authors declare no competing interests.

