## [Peer Review File · Nature Communications]

Reviewers' comments:

Reviewer #1 (Remarks to the Author):

Comments:

The authors designed and investigated a new D-A-D series of TADF emitters and applied their associated fluorescence and the delayed fluorescence emissions to construct a 3-D luminescent methodology. The 3D ratiometric luminescent sensor using TADF molecules may provide valuable information perhaps for both biological scientists and OLED researchers. However, the proposed mechanism here as well as the uniqueness of TADF molecules fitted to this mechanism is rather elusive, which should be gained much more in detail from both fundamental and applications points of view. At current stage, I cannot recommend this manuscript for publication in Nature Communication. Comments and suggestion are listed below:

1. Compound 3 shows dual emission in solution, the authors then assigned the emission band at 332 nm to a π - π^* transition LE emission. However, this LE emission band overlapped with the entire S1 absorption band (309-376 nm), which is impossible from the energy point of view because excitation at long wavelength cannot not gain short wavelength emission even plus thermal energy. For example, I have no idea how to have 376 nm excitation to give 332 nm emission. This unavoidably leads me to conclude that this emission band is either from S2 emission (extremely rare) or more plausibly the impurity emission. The authors should provide the excitation spectra of two emission bands and render a clear mechanism to explain the origin of dual/multiple emissions.
2. The authors indicated that the absorption band range from 309-376 nm is in the n - π^* character, which is forbidden based on optical transition. Any proof of this viewpoint? At least, the authors are encouraged to provide the absorption extinction coefficient and/or polarity/protic solvent dependence to prove that the absorption is n - π^* transition.
3. The authors used computed small T2-S1 energy gap to explain the TADF property is not from the large T1-S1 energy gap. However, the experimentally obtained ΔE_{S1T1} of 0.19 eV is not too large having the TADF property (ref. Nat. Photonics 2014, 8, 326-332.). Moreover, fundamentally, due to the rather small energy gap the internal conversion from T2 to T1 should be ultrafast ($k_{IC} > 10^{11} \text{ s}^{-1}$), which is expected to be much faster than the reversed intersystem crossing (RISC) calculated in this work. The authors have to give explanation why RISC is possible to occur at T2 state without relaxing to the T1 state. Note that even thermal energy can repopulate molecules from T1 to T2, $T2 \rightarrow S1$ RISC still cannot compete with the $T2 \rightarrow T1$ internal conversion.
4. The authors claim that calibration fitting was done under the air-saturated condition. TADF is drastically quenched by oxygen. In this regard, why authors used lifetime as one of the dimensions, instead of taking advantage of oxygen-sensitive property. Note that current 3-D sensor described in this manuscript can be established as well via common CT compound.
5. Authors indicated that solvatochromism is a key factor for the CT properties, and the authors provided the different polarity measurements of compound 1 and 2 in Fig. S2. However, data for compound 3 are not included in the SI or text, whereas authors used compound 3 for the illustrating 3-D ratiometric sensing. Authors should provide the different polarity measurements of 3 for comparisons.
6. Authors should provide comprehensive evidence to verify the TADF properties such as the difference between steady state PL measurements in oxygen free and aerated condition, time-resolved PL spectra and the oxygen free transient PL decay spectra, etc., rather than based on the temperature-dependent PL decay spectra and the theoretical calculation.
7. Table S3 shows the geometric difference between the S1 and T1 states based on the B3LYP Density Functional level. However, recent progresses (for example, Chem. Mater. 2017, 29, 477-478) indicate that B3LYP level opposes concept of typical Hartree-Fock methods and usually overlocalizes the wave functions. As a result, when it comes to the extended π -conjugated systems, B3LYP level may overestimate the energy barriers among the stable conformers. In other words, the calculated results using B3LYP level may underestimate the structure relaxation in consequence, which may not be suitable for the structural investigations of the titled compounds.

Since B3LYP level is used to conduct calculation on charge-transfer molecules here the authors have to convince the readers the suitability of B3LYP method in this study.

8. Authors used linear-fit curves to fit the 3-D plot diagram. Albeit the impressive results, authors have to provide theoretical background or any references to explain the choice of plots, i.e., the log wavelength ratios as x-axis, polarity as y-axis and the log lifetime ratio as z-axis. Do the slopes or the intercepts in the fitting line provide any physical meaning?

9. The authors conducted polarity sensing assay in different organic solvents and the complex PLs systems with different Chol contents and Hela cells. The 3-D ratiometric sensing method seems to provide more accurate measurements than other 2-D ratiometric sensing methods. However, lifetime of TADF is not only influenced by polarity, but also temperature, viscosity and oxygen concentrations. In vivo, the spatial dependent viscosity and hence diffusion rate and oxygen concentration dependence can dramatically change the lifetime of TADF in different organelles. These interfering factors are very important and I am surprised that authors did not have any consideration regarding these in this 3-D ratiometric sensing method. How do the authors prevent the errors/uncertainty caused by the influences of these parameters?

Reviewer #2 (Remarks to the Author):

In this manuscript, the authors designed and synthesized a donor-acceptor-donor molecule with dual emission. By using reasonable molecular engineering, one TADF molecule with charge-transfer singlet excited state and hybrid triplet excited states has been obtained. Since both of the wavelength and lifetime of TADF emission are correlated to polarity and can be easily altered by external environments, the TADF signal is served as a sensing signal. Correspondingly, the FL section always kept unchanged, so it works as an internal reference. By virtue of this serviceable trait of the TADF molecule, one 3-D working curve upon the ratiometric wavelength (Y-axis) and lifetime (Z-axis) versus polarity (X-axis) was established and applied into practical application both in vitro and in vivo. However, the mechanism of PL is lack of evidence and the application is just superficial, many supplementary experiments should be done. I suggest the manuscript should be reconsidered after major revision.

I have many comments that I would like to see addressed.

1. In the third paragraph, authors declared that "TADF emitters have been regarded as the third class OLED materials, as their metal-free long lifetime emission facilitates smart molecular engineering as well as the improvement of internal quantum efficiency." These statements are ambiguous and confusing. The lifetime of TADF molecules is generally relatively short (microsecond level), however, they still show excellent PL and EL performance in some cases. So it should be necessary to review the literature (such as *Nature Reviews Materials* 3, 18020 (2018)) and rewrite this part.

2. Also in the third paragraph, authors claimed that "..., it could be noted that the energy difference between the singlet and triplet states (ΔE_{ST}) can be minimized by adjusting the energy levels of the charge-transfer singlet state (1CT) state and the lowest locally excited triplet state (3LE)." It should be reminded that in some cases the triplet states of TADF molecules with high efficiency are dominated by CT nature but not LE, so this statement should be rewritten scrupulously.

3. The mechanism of dual emission for molecule 3 is unclear. The power-dependent emission spectra of complexes should be done to confirm TADF or TTA.

4. The authors calculated ΔE_{ST} from the peak difference between TADF (401 nm) and phosphorescence (427 nm), which is not accurate. In general, ΔE_{ST} is estimated experimentally from the onsets of the fluorescence and phosphorescence spectra for the 1CT and 3CT states.

5. From the results of HOMO and LUMO of compounds 2 and 3 shown in Figure 2, there is no conspicuous difference observed between them. But actually, only compound 3 exhibits distinct hybrid LE and CT nature, and dual emission. On the contrary, the compound 1 exhibits totally different MOs distribution with compounds 2 and 3. Nevertheless, the compound 1 shows similar PL performance with compound 2. Is the conjugated length of the HOMO distribution the only determining factor? This should be explained in detail.
6. From the NTOs results, the S1 state of compound 3 exhibits almost all CT nature as usual. But this phenomenon is relatively contradictory with obvious PL emission from 1LE as depicted in the manuscript. Please give reasonable explanation.
7. According to the application of the 3-D ratiometric luminescent sensing strategy in simulated membranes, it can be concluded that with the increasing content of Chol in the simulated membranes, the wavelength of TADF exhibits red-shift firstly, and then it shows discernible blue-shift. The authors attributed this blue-shift to redundantly exposed Chol environment. This explanation is stuffless, and this result need further evidence and explanation.
8. In page 6, para. 1: "The insensitivity of the signal to oxygen in our case might be due to the small T1→T2 gap (0.11 eV) and T2→S1 gap (0.12 eV), leading to an easier achievement of RISC to produce TADF." The reviewer thinks this statement may be not accurate. We guess the insensitivity of the signal to oxygen might be caused by the fact that phospholipids may effectively hinder the contact and collisions between the TADF molecule and oxygen molecules.
9. The PL spectrum of compound 3 in simulated membranes might be supposed to provide in the SI. In addition, PLQY of three molecules should be measured, especially, molecule 3, which is vital for photoluminescent imaging.
10. Photo-stability of molecule 3 should also be provided. Is there photobleaching when using for long-term imaging? The related experiments should be added.
11. The good resolution and high sensitivity is of importance for luminescent sensing. However, in this manuscript, the authors only changed three different Chol contents for sensing in simulated membranes. However, there is big polarity difference between them. Much more contents of Chol should be carried out to reduce the polarity difference and to detect the resolution. Additionally, the sensitivity is also designed to characterize and determine.
12. As a blue material, it is unsuitable for detecting in vivo due to cell damage by high energy blue light emission.
13. The formats of references are not uniform and there are several mistakes. Please check it again.

Reviewer #3 (Remarks to the Author):

The authors demonstrated a very interesting dual emission approach of LE-state normal fluorescence and CT-state thermally activated delayed fluorescence in a single molecule. This is uncommon in the extensively reported TADF emitters in OLEDs. The two emission mechanisms were well explained both through the theoretical and experimental methods. By utilizing this unique emission behavior, this work established an effective and stable 3-D ratiometric luminescent sensing method, since the lower-energy TADF emission wavelength and lifetime both changed with the solvent polarity, while the high-energy LE-state fluorescence as an internal reference remained unchanged. The strategy was further applied into a precise detection of the

microenvironmental polarity variation in complex phospholipid systems both in vitro and in vivo. The method provided new insights for convenient and accurate diagnosis of membrane lesions. The work may open new avenue for the design of new dual emission materials as well as the biological applications. I would like to suggest the acceptance of the work for publishing in Nature Communications after minor revision.

1 As new compounds, the elemental analysis of the three compounds should be provided to further verify the purity.

2 Since TADF emission concerns triplet CT excited state, the common TADF emitters are very sensitive to oxygen, with changed micro-second scale PL decay and PL quantum yield. Although the authors mentioned "The insensitivity of the signal to oxygen in our case might be due to the small T₁→T₂ gap (0.11 eV) and T₂→S₁ gap (0.12 eV)", it is still necessary to provide the PLQY at least in one solvent, such as DMF at ambient and degassed conditions. Besides, the ambient and degassed PL spectra for compound 3 in various solvents to confirm the TADF emission sensitivity to oxygen are needed to provide in supporting information

3 Some format problems, Page 1 Para. 1, last line, T₁ 1 should be subscript;

Figure 3 caption, 100 K to 300K, then page 5 Line 2, 77K, space missing;

Page 8, H NMR, J should be italic; same page, Para. 2 and 3, ¹H NMR and ¹³C NMR, 1 and 13 superscript.

Response to Referee 1's Comments

Comments: *The authors designed and investigated a new D-A-D series of TADF emitters and applied their associated fluorescence and the delayed fluorescence emissions to construct a 3-D luminescent methodology. The 3D ratiometric luminescent sensor using TADF molecules may provide valuable information perhaps for both biological scientists and OLED researchers. However, the proposed mechanism here as well as the uniqueness of TADF molecules fitted to this mechanism is rather elusive, which should be gained much more in detail from both fundamental and applications points of view. At current stage, I cannot recommend this manuscript for publication in Nature Communication.*

Our response: Thank you very much for the comments. Accordingly, we have done additional experiments to explain the proposed mechanism here as well as the uniqueness of the TADF molecules. Hopefully it could make the presented mechanism more clear. See comments and questions addressed below.

Question 1: *Compound 3 shows dual emission in solution, the authors then assigned the emission band at 332 nm to a π - π^* transition LE emission. However, this LE emission band overlapped with the entire S1 absorption band (309-376 nm), which is impossible from the energy point of view because excitation at long wavelength cannot not gain short wavelength emission even plus thermal energy. For example, I have no idea how to have 376 nm excitation to give 332 nm emission. This unavoidably leads me to conclude that this emission band is either from S2 emission (extremely rare) or more plausibly the impurity emission. The authors should provide the excitation spectra of two emission bands and render a clear mechanism to explain the origin of dual/multiple emissions.*

Answer 1: As the referee says, Compound 3 shows dual emission in solution, and we assigned the emission band at 332 nm to a π - π^* transition LE emission. However, we claimed that the LE absorption band is at 284 nm, which can be seen from the UV/Vis absorption spectra in Fig. 2(c) in the text. In order to make the results easier to understand, we changed the sentences "It shows analogous UV-Vis spectra in DCM and toluene with a strong absorption peak at 284 nm. This peak can be attributed to the π - π^* transitions of the AMID and the DPS units," in the third paragraph of **Results and Discussion** to be "The UV-Vis absorption spectra in toluene and DCM show a strong absorption peak at 284 nm, which can be attributed to the π - π^* transitions of the AMID,²³ and a broad absorption band at 309~376 nm." Please reference **Answer 2** for the details about "a broad absorption band at 309~376 nm".

According to the reviewer's suggestion, we provide the excitation spectra of 3 from the two emission bands at 332 nm and 435 nm (see Fig. S3d). As shown in Fig. S3d, the LE excitation band can be seen at 260~310 nm from the emission at 332 nm, while the ICT excitation band 310~370 nm is shown from the emission at 435 nm. These results are consistent with the UV-vis absorption spectra. We added a sentence in the third paragraph of the **Results and Discussion** section: "These results can also be confirmed by the excitation spectra (Fig. S3d) in DCM, in which the LE excitation band can be seen at 260~310 nm from the emission at 332 nm, while the ICT excitation band 310~370 nm is shown from the emission at 435 nm." In addition, what we should emphasize is that the compound 3 is pure from the ¹H NMR and HPLC (Fig. S13) and that there was no intermolecular interaction or aggregation effect of our molecule in DCM (2 μ M) as no dynamic light scattering (DLS) signal can be collected.

Question 2: *The authors indicated that the absorption band range from 309-376 nm is in the n- π^* character, which is forbidden based on optical transition. Any proof of this viewpoint? At least, the authors are encouraged to provide the absorption extinction coefficient and/or polarity/protic solvent dependence to prove that the absorption is n- π^* transition.*

Answer 2: In the third paragraph of **Results and Discussion**, we mentioned that “a weak absorption band at 309~376 nm caused by overlap of the n- π^* transition of AMID and a weak intramolecular charge transfer transition from AMID to DPS.” and “The other emission is a broad band with a maximum at 401 nm, which can be ascribed to the intramolecular charge-transfer (CT) transition because of their dipolar nature.” What mean here that the emission of 401 nm is ascribed to the intramolecular charge-transfer (ICT) transition and the absorption band of ICT transition is at 309~376 nm.

According to the reviewer’s suggestion, we provide the absorption extinction coefficient (Fig. S3a and Fig. S3b) and the polarity solvent dependence (Fig. S3c) of the band at 309~376 nm. The absorption extinction coefficient is $1.7 \times 10^4 \text{ M}^{-1} \cdot \text{cm}^{-1}$, so the band at 309~376 nm can be explained to one resulting from an efficient intermolecular charge transfer between donor units and acceptor units. The absorption behaves similarly to the CT properties of some TADF molecules in reported literature (*Adv. Mater.* 2017, 29, 1702767). We have cited this paper as **Reference 15**.

In order to make the results clearer to understand, we changed the sentences, “It shows analogous UV-Vis spectra in DCM and toluene with a strong absorption peak at 284 nm. This peak can be attributed to the π - π^* transitions of the AMID and the DPS units, and a weak absorption band at 309~376 nm caused by overlap of the n- π^* transition of AMID and a weak intramolecular charge transfer transition from AMID to DPS.”, in the third paragraph of **Results and Discussion** to be “The UV-Vis absorption spectra in toluene and DCM show a strong absorption peak at 284 nm, which can be attributed to the π - π^* transitions of the AMID, and a broad absorption band at 309~376 nm. The absorption extinction coefficient (Fig. S3a and Fig. S3b) of the band at 309~376 nm is $1.7 \times 10^4 \text{ M}^{-1} \cdot \text{cm}^{-1}$, resulting from an efficient intermolecular charge transfer between donor units and acceptor units. The absorption behaves similarly to the CT properties of some TADF molecules reported in literature 15 (Fig. S3c).”

Question 3: *The authors used computed small T_2 - S_1 energy gap to explain the TADF property is not from the large T_1 - S_1 energy gap. However, the experimentally obtained $\Delta E_{S_1T_1}$ of 0.19 eV is not too large having the TADF property (ref. *Nat. Photonics* 2014, 8, 326-332.). Moreover, fundamentally, due to the rather small energy gap the internal conversion from T_2 to T_1 should be ultrafast ($k_{IC} > 10^{11} \text{ s}^{-1}$), which is expected to be much faster than the reversed intersystem crossing (RISC) calculated in this work. The authors have to give explanation why RISC is possible to occur at T_2 state without relaxing to the T_1 state. Note that even thermal energy can repopulate molecules form T_1 to T_2 , $T_2 \rightarrow S_1$ RISC still cannot compete with the $T_2 \rightarrow T_1$ internal conversion.*

Answer 3: The role of T_2 state in the RICS process is still a challenge in studies of organic TADF materials. One of the possible mechanisms how upper T_n states can contribute the RICS rate was proposed by Penfold et. al. (*Chem. Phys. Chem.*, 2016, 5, 2956). They demonstrated that vibronic (nonadiabatic) coupling between the lowest local triplet (^3LE) state and lowest charge transfer triplet (^3CT) opens the possibility for significant second-order coupling effects and

increases krISC by about four orders of magnitude. This mechanism means that the vibronic coupling between the ³CT and ¹CT states using the ³LE state as an intermediate second-order term (see Eq. 7 in this reference).

$$k_{\text{rISC}} = \frac{2\pi}{\hbar} \left| \frac{\langle \Psi_{^1\text{CT}} | \hat{H}_{\text{SOC}} | \Psi_{^3\text{LE}} \rangle \langle \Psi_{^3\text{LE}} | \hat{H}_{\text{vib}} | \Psi_{^3\text{CT}} \rangle}{E_{^3\text{CT}} - E_{^3\text{LE}}} \right|^2 \delta(E_{^1\text{CT}} - E_{^3\text{LE}}) \quad (7)$$

At the same time the Spin orbit coupling (SOC term) in Eq. 7 is always large enough to promote RISC because of the different orbital symmetry of ³LE and ¹CT states. Moreover, the smaller the energy difference between ³LE and ³CT states in the denominator is the higher the RICS rate constant. We can expect this mechanism in our case because of the T₂ state for compound 3 contains a significant contribution of ³LE nature and thus the second-order coupling effects can really promote the increase of k_{RISC}.

On the other hand, there are some examples published in the literature of the last years that show the direct participation of T₂ and even higher T_n states in the RICS mediation (*J. Phys. Chem. C*, 2018, 122, 23934; *Chem. Phys. Chem.*, 2017, 18, 2314; *Organ. Electron.* 2018, 59, 45). Particularly, we want to stress the very recent publication by Monkman et.al. (*J. Phys. Chem. C* 2018, 122, 23934) who studied TADF of the well-known TCA blue emitter for which the S₁-T₁ gap is quite large (0.21 eV vs 0.17 eV for our case). Monkman and co-authors demonstrated that TADF of the TCA molecule arises through the mechanism of reverse internal conversion (rICDF). For this mechanism to be efficient, they have shown that the lowest triplet states T₂ and T₃ of TCA are very close in energy to S₁, facilitating efficient rISC. They have postulated that reverse internal conversion (rIC) between the triplet states gives rise to the upper triplet state population at room temperature and thus efficient rISC to S₁.

Using the Boltzmann distribution, we have obtained a 70:1 ratio between the population of T₁ and T₂ at room temperature for compound 3 (T₁-T₂ gap is 0.11 eV). It means that even if internal conversion from T₂ to T₁ is ultrafast, the population of T₂ at room temperature is statistically non-zero that provides the subsequent efficient T₂-S₁ RISC. The latter process is much more efficient comparing with the T₁-S₁ RISC because of 1) smaller energy gap and 2) admixture of LE configuration in T₂ state that provides efficient SOC between the T₂ and S₁. We have included a brief discussion of rIC and second-order coupling mediated mechanisms of DF for compound 3 into the main text. That is the sentences, “Moreover, the excitons in T2 are then able to convert to the 1CT state by the RISC process and radiatively decay to the ground state as TADF. The T₂↔T₁→S₁ RISC is much more efficient comparing with the T₁-S₁ RISC because of 1) a smaller S₁-T₂ energy gap and 2) admixture of LE configuration in the T₂ state that provides efficient SOC between T₂ and S₁. Very recently, Monkman et. al. experimentally proved such mechanism of reverse internal conversion delayed fluorescence (rICDF) for the well-known TCA blue emitter with quite large S₁-T₁ gap (0.21 eV vs 0.17 eV for our case). They postulated that reverse internal conversion (rIC) between the T₁ and T₂/T₃ triplet states gives rise to the upper triplet state population at room temperature, thus promoting an efficient rISC to S₁. Also, we can except in our case a strong RISC mechanism through the vibronic (nonadiabatic) coupling between the T₂ state, which contains the admixture of 3LE configuration, and the lowest charge transfer triplet state (3CT). Such second-order coupling effects can also enhance the kRISC rate together with the RISC through the rIC step.”, in the third paragraph of **TADF characteristic**.

Question 4: *The authors claim that calibration fitting was done under the air-saturated condition. TADF is drastically quenched by oxygen. In this regard, why authors used lifetime as one of the dimensions, instead of taking advantage of oxygen-sensitive property? Note that current 3-D sensor described in this manuscript can be established as well via common CT compound.*

Answer 4: The 3-D sensor described in this manuscript was established upon the ratiometric wavelength (Y-axis) and lifetime (Z-axis) versus polarity (X-axis). A common CT compound cannot do it due to the lack of an internal reference.

As the referee said, oxygen can affect the TADF property. However, the oxygen cannot be an orthogonal factor to the solvent polarity. So it is impossible for the oxygen to replace the lifetime as one of the dimensions. In addition, it is necessary for us here to keep oxygen for practical usage since it is hard to completely avoid it while the probe is applied in biosamples or even in vivo, as exemplified also in our work. As not all of the TADF signal was quenched in our case, the total demonstration could still be reasonable.

Question 5: Authors indicated that solvatochromism is a key factor for the CT properties, and the authors provided the different polarity measurements of compound 1 and 2 in Fig. S2. However, data for compound 3 are not included in the SI or text, whereas authors used compound 3 for the illustrating 3-D ratiometric sensing. Authors should provide the different polarity measurements of 3 for comparisons.

Answer 5: Actually, the emission spectra of compound 3 in different polarity solvents have already been shown in Fig. 4a. According to the reviewer's suggestion, we added the emission spectra of compound 3 in different polarity solvents in the Fig. S2c in order to well compare with the emission spectra of compound 1 and 2.

Question 6: Authors should provide comprehensive evidence to verify the TADF properties such as the difference between steady state PL measurements in oxygen free and aerated condition, time-resolved PL spectra and the oxygen free transient PL decay spectra, etc., rather than based on the temperature-dependent PL decay spectra and the theoretical calculation.

Answer 6: According to the reviewer's comments, we had provided comprehensive evidence to verify the TADF properties. Fig. 2c shows the time-resolved PL spectra of compound 3 in toluene at 77 K and we added some sentences in the 3th paragraph of **Design and Construction of the donor-acceptor-donor systems** of **Results and Discussion** section. That is, "The PL spectra also show dual emission at 332 nm and 427 nm at 77 K (see Fig. 2c). The emission around 332 nm disappears in the time-resolved photoluminescence spectra with a 100 μ s delay, while the emission around 427 nm shows a characteristic triplet excited state."

Fig. 3a shows the Transient PL decay and steady state PL emission spectra of compound 3 in oxygen free and saturated air conditions in toluene at 300 K. We added some sentences in the first paragraph of **TADF characteristic** of **Results and Discussion** section. That is, "In view of the long-lived T_1 excited state has been verified to exist, next we aim to clarify whether the emission at 401 nm in toluene is the TADF. We measured the steady state PL spectra and transient PL decay spectra in oxygen free and saturated air condition in toluene (Fig. 3a). The luminescence intensity at 401 nm measured under N_2 atmosphere is much higher than that measured under saturated air conditions. Moreover, the lifetime of 401 nm under air atmosphere decreased from 167 μ s, in oxygen-free environment environment, to 55 μ s. Thus, we can propose that the emission at 401 nm may be TADF from the 1CT state, which is obtained through a RISC process from the oxygen-sensitive triplet state."

Table S6 shows the PLQY of compound 3 in oxygen free and saturated air conditions in different solvents. The PLQY under oxygen free is higher than it under saturated air conditions.

The results indicate that oxygen can quench TADF to some extent, but not completely. Therefore, the ratiometric photoluminescence sensing strategy demonstrated in this work is still valid and useful. We changed the sentences “Another point worth mentioning is that all these wavelengths and lifetimes of the TADF band were investigated in an air-saturated environment, which is sufficient for the establishment and application of the 3-D plot diagram.” to “Another point worth mentioning is that all these wavelengths and lifetimes of the TADF band were investigated in an air-saturated environment, which is sufficient for the establishment and application of the 3-D plot diagram from the PLQY results (Table S6). This is because that oxygen can quench TADF to some extent (Fig. S7), but not completely. Therefore, the ratiometric photoluminescence sensing strategy demonstrated in this work is still valid and useful.” in the third paragraph of **3-D ratiometrically luminescent sensing** of **Results and Discussion** section.

***Question 7:** Table S3 shows the geometric difference between the S_1 and T_1 states based on the B3LYP Density Functional level. However, recent progresses (for example, Chem. Mater. 2017, 29, 477-478) indicate that B3LYP level opposes concept of typical Hartree-Fock methods and usually overlocalizes the wave functions. As a result, when it comes to the extended -conjugated systems, B3LYP level may overestimate the energy barriers among the stable conformers. In other words, the calculated results using B3LYP level may underestimate the structure relaxation in consequence, which may not be suitable for the structural investigations of the titled compounds. Since B3LYP level is used to conduct calculation on charge-transfer molecules here the authors have to convince the readers the suitability of B3LYP method in this study.*

Answer 7: The Referee is absolutely right that there are some limitations of the usage of B3LYP level of theory for optimizing of conformational structure of pi-conjugated systems. Especially, the “classical” B3LYP fails to well predict the energy of charge-transfer excited states. To avoid this point, we tuned the part of non-local Hartree-Fock exchange from the 20% to 12%. The further decreasing provides a transformation of CT to LE configuration for the S_1 and T_1 states. However, the conformational structure of the ground (S_0) and relaxed S_1 and T_1 electronic states remains almost the same (deviations no more 1 °) upon the variation of the HF exchange term. We have also tested the dispersion-containing wB97XD functional and long-range corrected CAM-B3LYP functional for the optimization of the studied systems and subsequent TDDFT calculations. We found that there are no significant differences in the structures of the S_0 , S_1 and T_1 states comparing with the B3LYP. The dihedral angles between the D and A moieties are still almost the same (deviation no more than 5 °). At the same time, the energies of the excited states were found to be strongly overestimated comparing with the parameterized B3LYP functional ($\alpha_X^{\text{HF}}=0.12$, $\alpha_X^{\text{Slater}}=0.88$). That is why we presented in the manuscript only the results based on the parameterized B3LYP/6-31+G(d) approximation (PCM solvent model was included). We have added sentences, “We have also tested the dispersion-containing wB97XD functional and long-range corrected CAM-B3LYP functional for the optimization of the studied systems and the subsequent TDDFT calculations. We have found that there is no significant difference in the structure of S_0 , S_1 and T_1 states comparing with the B3LYP results. At the same time, the energies of excited states were found to be strongly overestimated comparing with the parameterized B3LYP functional.” in the **Computational details** of the **Methods** section.

***Question 8:** Authors used linear-fit curves to fit the 3-D plot diagram. Albeit the impressive results, authors have to provide theoretical background or any references to explain the choice of plots, i.e., the log wavelength ratios as x-axis, polarity as y-axis and the log lifetime ratio as*

z-axis. Do the slopes or the intercepts in the fitting line provide any physical meaning?

Answer 8: In this paper, we established the 3-D working curve upon the ratiometric wavelength (Y-axis) and lifetime (Z-axis) versus polarity (X-axis). At first, we found compound 3 possesses a large solvatochromic (Fig. 4a and Fig. S2c), which results from the difference in dipole moments between the excited and the ground states. According to the Lippert-Mataga equation, the Stokes shift / peak / ratiometric of the emission is versus the solvent polarity (*J. Am. Chem. Soc.* 2005, 127, 1300 (**Reference 22**)). The reason we assumed log wavelength ratios as Y-axis is that an excellent linear relationship ($R^2 = 0.99$) exists between the log wavelength ratios and polarity. Then, for consistency, we employed log lifetime ratios as Z-axis, which also shows a good linear relationship ($R^2 = 0.95$) with the polarity. We added the fitting data in Fig. S4. We consider the slope in the fitting line shows the rate of log wavelength ratios or log lifetime ratios changes with polarity. However, the intercepts in the fitting line cannot provide any physical meaning because the 3-D working curve is established and applied in systems with polarity, namely that it does not mean anything when the polarity is 0.

We have added the fitting results “($R^2 = 0.99$)” in the first paragraph and “($R^2 = 0.95$)” in the second paragraph of **3-D ratiometrically luminescent sensing** of **Results and Discussion** section. In addition, the literatures: *J. Am. Chem. Soc.* 2005, 127, 1300 had been cited as **Reference 22**.

Question 9: *The authors conducted polarity sensing assay in different organic solvents and the complex PLs systems with different Chol contents and Hela cells. The 3-D ratiometric sensing method seems to provide more accurate measurements than other 2-D ratiometric sensing methods. However, lifetime of TADF is not only influenced by polarity, but also temperature, viscosity and oxygen concentrations. In vivo, the spatial dependent viscosity and hence diffusion rate and oxygen concentration dependence can dramatically change the lifetime of TADF in different organelles. These interfering factors are very important and I am surprised that authors did not have any consideration regarding these in this 3-D ratiometric sensing method. How do the authors prevent the errors/uncertainty caused by the influences of these parameters?*

Answer 9: We also approve the Referee’s comments that the lifetime of TADF is not only influenced by polarity, but also the temperature, viscosity and oxygen concentrations. We here established the 3-D ratiometric sensing method in different solvents and applied it to simulated membranes. For the different solvents and simulated membranes, the temperature (room temperature) and oxygen concentrations (air-saturated) are basically at the same level. Then we studied the effects of different viscosity on steady state PL and transient PL decay spectra of compound 3 by adding different glycerol concentrations to the ethanol. The results have been shown in Fig. S6 and indicated that although the intensity of FL and TADF changed along with the increase of glycerol concentrations, the wavelength and lifetime varied little. It means that viscosity did not affect the establishment of the 3-D ratiometrically luminescent sensing like this. We have added some sentences in the third paragraph of **3-D ratiometrically luminescent sensing** of **Results and Discussion** section: “We studied the effects of different viscosity on steady state PL and transient PL decay spectra of compound 3 by adding different glycerol concentrations to the ethanol. The results (Fig. S6) show that although the intensity of the dual band can be affected with the increase of the glycerol concentrations, the wavelength and lifetime varies only little. It suggests that the choice of wavelength and lifetime as the dimensions is practical.”

Response to Referee 2's Comments

Comments: *In this manuscript, the authors designed and synthesized a donor–acceptor–donor molecule with dual emission. By using reasonable molecular engineering, one TADF molecule with charge-transfer singlet excited state and hybrid triplet excited states has been obtained. Since both of the wavelength and lifetime of TADF emission are correlated to polarity and can be easily altered by external environments, the TADF signal is served as a sensing signal. Correspondingly, the FL section always kept unchanged, so it works as an internal reference. By virtue of this serviceable trait of the TADF molecule, one 3-D working curve upon the ratiometric wavelength (Y-axis) and lifetime (Z-axis) versus polarity (X-axis) was established and applied into practical application both in vitro and in vivo. However, the mechanism of PL is lack of evidence and the application is just superficial, many supplementary experiments should be done. I suggest the manuscript should reconsidered after major revision.*

Our response: Thanks for your positive comments and valuable suggestions. Accordingly, we did a large amount of fundamental experiments to explain the proposed mechanism here as well as the uniqueness of the TADF molecules. Moreover, we carried out measurements to explain the application of 3-D ratiometrically luminescent sensing abundantly. Hopefully these results could make the presented mechanism and application more clear now. See comments and questions addressed below.

Question 1: *In the third paragraph, authors declared that “TADF emitters have been regarded as the third class OLED materials, as their metal-free long lifetime emission facilitates smart molecular engineering as well as the improvement of internal quantum efficiency.” These statements are ambiguous and confusing. The lifetime of TADF molecules is generally relatively short (microsecond level), however, they still shows excellent PL and EL performance in some case. So it should be necessary to review the literature (such as *Nature Reviews Materials* 3, 18020 (2018)) and rewrite this part.*

Answer 1: The sentence, “TADF emitters have been regarded as the third class OLED materials, as their metal-free long lifetime emission facilitates smart molecular engineering as well as the improvement of internal quantum efficiency.”, in the third paragraph has been changed into the one, “TADF emitters have been regarded as the third class of OLED materials, as their harvest of both singlet and triplet excitons without noble metals facilitates smart molecular engineering with an improved internal quantum efficiency.” We cited the literature (*Nature Reviews Materials* 2018, 3, 18020) as **References** 14.

Question 2: *Also in the third paragraph, authors claimed that “..., it could be noted that the energy difference between the singlet and triplet states (ΔE_{ST}) can be minimized by adjusting the energy levels of the charge-transfer singlet state (1CT) state and the lowest locally excited triplet state (3LE).” It should be reminded that in some cases the triplet states of TADF molecules with high efficiency are dominated by CT nature but not LE, so this statement should be rewritten scrupulously.*

Answer 2: The sentence, “..., it could be noted that the energy difference between the singlet and triplet states (ΔE_{ST}) can be minimized by adjusting the energy levels of the charge-transfer singlet

state (^1CT) state and the lowest locally excited triplet state (^3LE).”, in the third paragraph has been changed into the one, “As reported for TADF molecules, the energy difference between the singlet and triplet states (ΔE_{ST}) can be minimized by adjusting the energy levels of the lowest locally excited triplet state (^3LE) and charge-transfer states (both ^1CT and ^3CT)”

Question 3: *The mechanism of dual emission for molecule 3 is unclear. The power-dependent emission spectra of complexes should be done to confirm TADF or TTA.*

Answer 3: As a power adjuster to provide the power-dependent emission spectra is usually not easy to access, we here used optical attenuators to adjust the intensity of excitation for this experiment. The spectra of intensity variation of emission with excitation transmittance are shown in Fig. 3c under oxygen free condition in DCM (delay 20 μs). A clear linear dependence of the intensity integral with excitation transmittance is observed (Fig. 3d), confirming the pure thermally assisted nature of the TADF mechanism in compound 3. We added some sentences in the first paragraph of **TADF characteristic** of **Results and Discussion** section. That is, “In addition, a strictly linear dependence of the intensity integral with excitation transmittance is observed (see Fig. 3c and Fig. 3d), confirming the pure thermally assisted nature of the TADF mechanism in compound 3.”

On the other hand, TTA is even impossible to be observed in this way here because TTA is an up-conversion emission process with the application of excitation with longer wavelength, which is totally different from our excitation case.

Question 4: *The authors calculated ΔE_{ST} from the peak difference between TADF (401 nm) and phosphorescence (427 nm), which is not accurate. In general, ΔE_{ST} is estimated experimentally from the onsets of the fluorescence and phosphorescence spectra for the ^1CT and ^3CT states.*

Answer 4: Thanks for your valuable suggestion. The ΔE_{ST} is corrected to be 0.17 eV, which is reestimated experimentally from the onsets of the fluorescence and phosphorescence spectra (Fig. 2c).

Question 5: *From the results of HOMO and LUMO of compounds 2 and 3 shown in Figure 2, there is no conspicuous difference observed between them. But actually, only compound 3 exhibits distinct hybrid LE and CT nature, and dual emission. On the contrary, the compound 1 exhibits totally different MOs distribution with compounds 2 and 3. Nevertheless, the compound 1 shows similar PL performance with compound 2. Is the conjugated length of the HOMO distribution the only determining factor? This should be explained in detail.*

Answer 5: The special spectroscopic behavior of compound 3 (if compared with compound 2) is caused by the significant contribution of the LE configuration into the T_1 and T_2 states. It promotes a significant second-order coupling between these states and also non-zero spin-orbit coupling between the ^1CT state and mixed ($^3\text{CT}+^3\text{LE}$) T_1 and T_2 states (see fig. 3e and Table S4). Please reference **Answer 3** to **Referee 1's comments** for details. On the other hand, for compound 2, which is very similar to compound 3, the T_1 and T_2 states are almost pure CT states and thus there is no efficient second-order vibronic coupling between them and no significant spin-orbit coupling between them and the S_1 (^1CT) state. Thus, for compound 2 we observe only the prompt fluorescence which is red-shifted relative to the compound 3 in agreement with TDDFT calculations (2.78 eV for compound 2 comparing with the 3.09 eV for compound 3,

Table S1). Compound 1 is a standalone system for which the S_1 , T_1 and T_2 states correspond purely to the LE nature. Moreover, the T_1 and T_2 states are degenerate and lie about 0.35 eV lower than S_1 , so no TADF is observed in this case. We added some sentences in the third paragraph of **TADF characteristic of Results and Discussion** section. That is, “For compound 2, the T_1 and T_2 states are almost pure CT states and thus there is no efficient second-order vibronic coupling between them and no significant spin-orbit coupling between them and the S_1 (1CT) state. Thus, for compound 2 we observe only the prompt fluorescence which is red-shifted relative to the compound 3 in agreement with TDDFT calculations (2.78 eV for compound 2 comparing with the 3.09 eV for compound 3, Table S1). Compound 1 is a standalone system for which the S_1 , T_1 and T_2 states correspond purely to the LE nature. Moreover, the T_1 and T_2 states are degenerate and lie about 0.35 eV lower than S_1 , so no TADF is observed in this case.”

Question 6: *From the NTOs results, the S_1 state of compound 3 exhibits almost all CT nature as usual. But this phenomenon is relatively contradictory with obvious PL emission from 1LE as depicted in the manuscript. Please give reasonable explanation.*

Answer 6: To explain this comment we should say at first that compound 3 represents a typical D-A-D structure. The D and A moieties are coupled by a single C-N bond and thus D and A fragments are rotated along this axis (look at the dihedral angles between donor and acceptor in Table S3). For such type of non-planar D-A-D systems the expected 1CT fluorescence is usually accompanied by the 1LE emission from the donor moieties (see *J. Phys. Chem. C* 2017, 121, 17764 for example for the similar D-A-D and D-A-D' systems). This literature had been cited as **Reference 16**.

Question 7: *According to the application of the 3-D ratiometric luminescent sensing strategy in simulated membranes, it can be concluded that with the increasing content of Chol in the simulated membranes, the wavelength of TADF exhibits red-shift firstly, and then it shows discernible blue-shift. The authors attributed this blue-shift to redundantly exposed Chol environment. This explanation is stuffless, and this result need further evidence and explanation.*

Answer 7: In fact, with the increasing contents of Chol in the simulated membranes, the wavelength of TADF exhibits blue-shift firstly, and then it shows a discernible red-shift. We utilized TEM and DLS measurement to trace nanostructures with different Chol contents to provide further evidence and explanation. The results are shown in Fig. S10. When the mass ratio of 3/PLs/Chol is 1/100/0, 1/100/60 and 1/100/100, the Z-average diameter was determined to be *ca.* 85 nm, 140 nm and 120 nm, respectively.

It is known that for the simulated membranes of PLs, the head group components, phosphate and choline, are found within the interface regions, along with water of hydration. Carbonyls and contiguous methylene groups from the alkyl chains are also associated with this region. Accordingly, the central section of the membrane is nearly isotropic. Without hydrophilicity, the compound 3 is dispersed in the central section and its environment polarity profile is essentially similar to the central section.

Without Chol (1/100/0 of 3/PLs/Chol) of the membrane, the polarity comes from the long alkyl chain of PLs (Fig. 5a (I)) and the Z-average diameter was determined to be *ca.* 85 nm.

When the mass ratio of 3/PLs/Chol is 1/100/60, the Chol molecules are arranged perpendicular to the surface, with the hydroxyl group positioned near the middle of the headgroup region of the PLs molecule (Fig. 5a (II)). The central part of the membrane consists of the long alkyl chain of

PLs and polycyclic aromatic hydrocarbon chain of Chol. So its polarity decreased and the wavelength of TADF blue shift accordingly. Moreover, the Z-average diameter was increased to be *ca.* 140 nm indicating more substances formed the membranes.

The polarity clearly increases for the membrane with 3/PLs/Chol of 1/100/100 because 1) the sensor largely exposes to Chol environment; 2) the presence of a higher proportion of Chol may induce chain ordering and close packing increase and a wider separation of the phospholipid headgroups, which may surround the sensor molecules. Moreover, the Z-average diameter was decreased to be *ca.* 120 nm, which is due to the condensing effect of Chol on PLs, by reducing average orientational and positional ordering of the alkyl chains and consequently reducing the free volume.

In fact, similar results that the polarity decreases in bilayers containing 35 mol% Chol and then increases with 45 mol% of Chol are reported in related literature (Biochim. Biophys. Acta. 2007, 1768, 2914). If we convert the mass ratio with PLs/Chol of 100/60 and 100/100 in our system to molar content they are 33.6 mol% Chol and 45.7 mol% Chol (The molar mass of PLs is 325 g/mol and the molar mass of Chol is 386 g/mol). We cited this report as Ref 33 to support our interpretation.

Therefore, we changed some sentences in the third paragraph of **Application of the 3-D ratiometric luminescent sensing strategy in simulated membranes** of **Results and Discussion** section accordingly (see the corresponding highlighted part in the text).

Question 8: In page 6, para. 1: “The insensitivity of the signal to oxygen in our case might be due to the small $T_1 \rightarrow T_2$ gap (0.11 eV) and $T_2 \rightarrow S_1$ gap (0.12 eV), leading to an easier achievement of RISC to produce TADF.” The reviewer thinks this statement may be not accurate. We guess the insensitivity of the signal to oxygen might be caused by the fact that phospholipids may effectively hinder the contact and collisions between the TADF molecule and oxygen molecules.

Answer 8: Thanks for your valuable suggestion. We highly approve the Referee’s comments that the insensitivity of the signal to oxygen might be caused by the fact that phospholipids may effectively hinder the contact and collisions between the TADF molecule and oxygen molecules. This factor makes our probe reliable in practical usage. However, for those fundamental studies, we still believe that our interpretation is reasonable because our compound has a second-order coupling effects to facilitate the $T_1 \rightarrow T_2$ and $T_2 \rightarrow S_1$ process (see Answer 3 to the Reviewer 1 for details).

We omitted the sentences “The insensitivity of the signal to oxygen in our case might be due to the small $T_1 \rightarrow T_2$ gap (0.11 eV) and $T_2 \rightarrow S_1$ gap (0.12 eV), leading to an easier achievement of RISC to produce TADF.³¹” in the third paragraph of **3-D ratiometrically luminescent sensing** of **Results and Discussion** section. So we added the sentences “The insensitivity of the signal to oxygen might be caused by the fact that phospholipids may effectively hinder the contact and collisions between the TADF molecule and oxygen molecules. This factor makes our probe reliable in practical usage.” in the last paragraph of **Application of the 3-D ratiometric luminescent sensing strategy in simulated membranes** of **Results and Discussion** section.

Question 9: The PL spectrum of compound 3 in simulated membranes might be supposed to provide in the SI. In addition, PLQY of three molecules should be measured, especially, molecule 3, which is vital for photoluminescent imaging.

Answer 9: The PL spectra and lifetime of compound 3 in simulated membranes had been provided in the Fig. S8 and Fig. S9 (see **Answer 11** for details). The PLQY of compounds 1 and 2 in DCM and compound 3 in different solvents had been measured and shown in Table S7 and Table S6.

Question 10: *Photo-stability of molecule 3 should also be provided. Is there photobleaching when using for long-term imaging? The related experiments should be added.*

Answer 10: Photo-stability of molecule 3 in toluene under 300 nm UV light within 120 min had been provided in Fig. S12.

Question 11: *The good resolution and high sensitivity is of importance for luminescent sensing. However, in this manuscript, the authors only changed three different Chol contents for sensing in simulated membranes. However, there is big polarity difference between them. Much more contents of Chol should be carried out to reduce the polarity difference and to detect the resolution. Additionally, the sensitivity is also designed to characterize and determine.*

Answer 11: We believe that the good resolution and high sensitivity is of importance for luminescent sensing. In fact, we designed 8 different Chol contents (1/100/0, 1/100/20, 1/100/40, 1/100/60, 1/100/80, 1/100/100, 1/100/150 and 1/100/200) for sensing in the simulated membranes. The PL spectra (Fig. S8 and Table S6) show that maximum emission wavelength of 1/100/0, 1/100/20, 1/100/40, 1/100/60, 1/100/80, 1/100/100, 1/100/150 and 1/100/200 are 457 nm, 448 nm, 446 nm, 445 nm, 452 nm, 466 nm, 466 nm and 466 nm, respectively. The maximum emission wavelength blue-shifted from 457 nm to 445 nm when the Chol contents increased from 1/100/0 to 1/100/60, and red-shifted from 445 nm to 466 nm when the Chol contents went on increasing from 1/100/60 to 1/100/100. It then kept stable when the Chol contents increased from 1/100/100 to 1/100/200. Moreover, the PL lifetime of every Chol contents was also measured and shown in Fig. S9 and Table S6. The lifetime of 1/100/0, 1/100/20, 1/100/40, 1/100/60, 1/100/80, 1/100/100, 1/100/150 and 1/100/200 are 4.12 μ s, 9.09 μ s, 9.09 μ s, 9.09 μ s, 7.86 μ s, 3.17 μ s, 3.17 μ s and 3.17 μ s, respectively. The lifetime increased from 4.12 μ s to 9.09 μ s when the Chol contents increased from 1/100/0 to 1/100/60, and decreased from 9.09 μ s to 7.86 μ s when the Chol contents went on increasing from 1/100/60 to 1/100/100, and then keep stable when the Chol contents increased from 1/100/100 to 1/100/200. Therefore, we divided the polarity microenvironment of compound 3 in the complex PLs system into three stages: 1) starting value with Chol contents of 1/100/0; 2) polarity decreased from 1/100/0 to 1/100/60; 3) polarity increased from 1/100/60 to 1/100/100. In the text, we analyzed three representative Chol content of 0, 60 and 100 (named A, B and C) in detail.

Therefore, we changed some sentences in the second paragraph of **Application of the 3-D ratiometric luminescent sensing strategy in simulated membranes** of **Results and Discussion** section accordingly (see the corresponding highlighted part in the text).

Question 12: *As a blue material, it is unsuitable for detecting in vivo due to cell damage by high energy blue light emission.*

Answer 12: The purpose of the cell imaging study is to show the visualization of our probe in practical usage, which is essentially connected to the microenvironmental detection of the membrane system. In fact, confocal fluorescence microscopy was carried out with the Hela cells

fixed by 4% paraformaldehyde and dead on culture dish. In addition, we believe that such a material could be acceptable since we just employed the blue light emission for detection, not for a long-term physiological activity.

Question 13: *The formats of references are not uniform and there are several mistakes. Please check it again.*

Answer 13: We carefully checked the formats of references again.

Response to Referee 3's Comments

Comments: *The authors demonstrated a very interesting dual emission approach of LE-state normal fluorescence and CT-state thermally activated delayed fluorescence in a single molecule. This is uncommon in the extensively reported TADF emitters in OLEDs. The two emission mechanisms were well explained both through the theoretical and experimental methods. By utilizing this unique emission behavior, this work established an effective and stable 3-D ratiometric luminescent sensing method, since the lower-energy TADF emission wavelength and lifetime both changed with the solvent polarity, while the high-energy LE-state fluorescence as an internal reference remained unchanged. The strategy was further applied into a precise detection of the microenvironmental polarity variation in complex phospholipid systems both in vitro and in vivo. The method provided new insights for convenient and accurate diagnosis of membrane lesions. The work may open new avenue for the design of new dual emission materials as well as the biological applications. I would like to suggest the acceptance of the work for publishing in Nature Communications after minor revision.*

Our response: Thanks for your positive comments. See comments and questions addressed below.

Question 1: *As new compounds, the elemental analysis of the three compounds should be provided to further verify the purity.*

Answer 1: According to the reviewer's comments, we further utilized the elemental analysis measurement and high performance liquid chromatography (HPLC) to verify the purity of the three compounds. Please see **Methods** and Fig. S13d for detailed information.

Question 2: *Since TADF emission concerns triplet CT excited state, the common TADF emitters are very sensitive to oxygen, with changed micro-second scale PL decay and PL quantum yield. Although the authors mentioned "The insensitivity of the signal to oxygen in our case might be due to the small T1→T2 gap (0.11 eV) and T2→S1 gap (0.12 eV)", it is still necessary to provide the PLQY at least in one solvent, such as DMF at ambient and degassed conditions. Besides, the ambient and degassed PL spectra for compound 3 in various solvents to confirm the TADF emission sensitivity to oxygen are needed to provide in supporting information*

Answer 2: According to the reviewer's comments, the PLQY and PL spectra under ambient and degassed condition for compound 3 in various solvents are provided. Please refer to Table S6 for the PLQY detailed information and Fig. S7 for PL spectra detailed information.

Question 3: *Some format problems, Page 1 Para. 1, last line, T1 1 should be subscript; Figure 3 caption, 100 K to 300K, then page 5 Line 2, 77K, space missing; Page 8, H NMR, J should be italic; same page, Para. 2 and 3, 1H NMR and 13C NMR, 1 and 13 superscript.*

Answer 3: According to the reviewer's comments, we checked the format carefully.

"T1" has been corrected to be "T₁";

Figure 3 caption, 100 K to 300K, "300K" has been corrected to be "300 K";

Page 5 Line 2, "77K" has been corrected to be "77 K";

Page 8, H NMR, we replaced all the “J” with “*J*”;

Same page, Para. 2 and 3, we replaced all “¹H NMR” with “¹H NMR” and “¹³C NMR” with “¹³C NMR”.

We also checked all the text and corrected those similar format issues.

REVIEWERS' COMMENTS:

Reviewer #2 (Remarks to the Author):

This paper presents some very interesting results on TADF molecules as a high-precision ratiometric sensor. The authors have fully addressed the comments from the reviewers. Thus, I highly recommend its publication in Nat. Commun. But a little bit of details could be further improved. Such as,

- 1) Some typing format problems, like an extra space of "to avoid" and "to overcome" in Para. 1, Page 1. The authors should check throughout the text again.
- 2) Details about the measurement by HPLC, delayed emissions spectra and PL quantum yield are needed to provide in Methods section.

Reviewer #3 (Remarks to the Author):

The authors have been carefully addressed all the questions from the reviewers. Sufficient evidence for the dual emission mechanism were provided, especially various PL measurements were done for TADF property. It is suitable for publication now.

Response to Referee 2's Comments

Comments: This paper presents some very interesting results on TADF molecules as a high-precision ratiometric sensor. The authors have fully addressed the comments from the reviewers. Thus, I highly recommend its publication in Nat. Commun. But a little bit of details could be further improved. Such as,

1) Some typing format problems, like an extra space of "to avoid" and "to overcome" in Para. 1, Page 1. The authors should check throughout the text again.

2) Details about the measurement by HPLC, delayed emissions spectra and PL quantum yield are needed to provide in Methods section..

Our response: We thank the reviewer for the support. We have corrected these issues and provided the required information accordingly (see the corrections in HIGHLIGHT font in the maintext).

Response to Referee 3's Comments

Comments: The authors have been carefully addressed all the questions from the reviewers. Sufficient evidence for the dual emission mechanism were provided, especially various PL measurements were done for TADF property. It is suitable for publication now

Our response: We thank the reviewer for the support.